# Lectotypifications of Three Names in *Garcinia*, Synonymy of *Garcinia pedunculata* and Detailed Descriptions of Three Species in *Garcinia* Section *Brindonia* (Clusiaceae)

Chatchai Ngernsaengsaruay [1,2] 

1   Department of Botany, Faculty of Science, Kasetsart University, Chatuchak, Bangkok 10900, Thailand; fsciccn@ku.ac.th; Tel.: +66-2562-5555 (ext. 646303 or 646317) or +66-81-914-0580
2   Center for Advanced Studies in Tropical Natural Resources (CASTNaR), National Research University-Kasetsart University (NRU-KU), Chatuchak, Bangkok 10900, Thailand

**Abstract:** A revision of the genus *Garcinia* has recently been undertaken by the author as part of the *Flora of Thailand*. Herbarium specimens deposited in several herbaria, and those included in the digital herbarium databases, were examined by consulting taxonomic literature. In this study, the three names in *Garcinia* section *Brindonia* are lectotypified as *G. gracilis*, *G. lanceifolia* and *G. planchonii*. A new synonym for *G. pedunculata*, namely *G. planchonii*, is proposed. Detailed descriptions, recognitions and illustrations of three species in *Garcinia* (*G. atroviridis*, *G. lanceifolia* and *G. pedunculata*) are presented, along with information on distributions, specimens examined, habitats and ecology, IUCN conservation status, phenology, etymology, vernacular names and uses. The fruits, the young shoots and leaves, and the flowers of these three species are edible and have a sour taste. These species are often cultivated for their fruits.

**Keywords:** black gland dots; colourless latex; edible plants; dioecious plant; *Garcinia atroviridis*; *Garcinia lanceifolia*; interrupted wavy lines; plant taxonomy; sour relish; yellow latex

## 1. Introduction

*Garcinia* L. is a group of dioecious, sometimes polygamo-dioecious, evergreen trees, occasionally shrubs and the largest genus in the *Clusiaceae Lindl.* (*Guttiferae Juss.*). This genus contains 404 accepted species and is distributed throughout the tropics and subtropics [1], with centres of diversity in Southeast Asia and Madagascar [2]. Previous studies on *Garcinia* revealed c. 60 species in the Malay Peninsula [3–6], 40 species in India [7–13], 34 species in Indo-China [14], 30 species in British India [15] and 20 species in China [16]. In Thailand, 20 species of the genus *Garcinia* were enumerated by Craib [17], the northern region had 6 species [18] and the peninsular region had 23 species (including 5 unidentified species) [19]. In 2016 and 2022, Ngernsaengsaruay & Suddee described new species, *G. nuntasaenii* and *G. santisukiana*, respectively, in Thailand [20,21].

*Garcinia* section *Brindonia* (Thouars) Choisy is characterised by the following: stamens in one central mass or column or in a ring (*Garcinia atroviridis*); anthers 4-thecous, ellipsoid or rectangular; pollen colporate, reticulate; stigma generally completely divided into the same number of rays as there are locules of the ovary, verrucose; sepals and petals 4; pistillode absent (except in *G. atroviridis*). This section is one of the largest and is also one of the best known because several species are cultivated for their edible fruits. The section is widespread in Southeast Asia, with its range extending from India and Sri Lanka through Indo-China as far as southern China and throughout Indonesia and the Philippines [22].

During the preparation of a recent taxonomic revision of the genus *Garcinia* as part of the *Flora of Thailand*, I identified three names (*G. gracilis*, *G. lanceifolia* and *G. planchonii*) in the section *Brindonia* that required lectotypifications following the ICN [23]. *G. planchonii* is treated here as a new synonym for *G. pedunculata*. Detailed descriptions, recognitions

and illustrations of three species in *Garcinia* (*G. atroviridis*, *G. lanceifolia* and *G. pedunculata*) are provided, along with information on distributions, specimens examined, habitats and ecology, IUCN conservation status, phenology, etymology, vernacular names and uses.

## 2. Materials and Methods

Herbarium specimens were deposited in AAU, BKF, BM, C, CMUB, K, K-W, P, PSU, and QBG, and those included in the digital herbarium databases of AAU [24], CAL [25], BM [26], E [27], G, G-DC [28], K, K-W [29], L, U [30] and P [31] were examined by consulting the taxonomic literature [3–7,12,14–19,22,25,32–54]. All acronyms follow Thiers [55]. All specimens cited have been seen by the author, unless stated otherwise. The taxonomic history of three species in the *Garcinia* section *Brindonia* was compiled using the taxonomic literature and by consulting online databases [1,56]. The morphological characters, distributions, habitats, phenology and uses were described from historic and newly collected herbarium specimens and from the author's observations during the fieldwork. The vernacular names were compiled from the specimens examined and the literature [17,19,57]. The conservation status of each species was assessed following the IUCN Red List Categories and Criteria [58] for a preliminary assessment of the conservation category in combination with GeoCAT analysis [59] and field information. The calculations of the Extent of Occurrence (EOO) and Area of Occupancy (AOO) were based on GeoCAT [60].

## 3. Results

*3.1. Taxonomic Treatment*

3.1.1. *Garcinia atroviridis* Griff. ex T. Anderson (Anderson, 1874)

*Garcinia atroviridis* Griff. ex T. Anderson [3]: 159; [4]: 272; [5]: 173; [6]: 206; [7]: 118, t. 1, fig. 10; [12]: 105; [15]: 266; [17]: 114; [19]: 348, fig. 535; [22]: 328; [32]: 50; [34]: 314, fig. 102; [35]: 186; [39]: 109; [40]: 48; [49]: 260; [50]: 176. Type: India, Upper Assam, Tabong, female fl., very y. fr., s.d., *Griffith 862* (Herbarium Hookerianum 1867, Distributed at the Royal Gardens, Kew 1861–2, lectotype K [K000677601!], designated by Maheshwari [7]) (Figure 1).

*Description.* Evergreen trees, 6–20 m tall, 40–100 cm girth; latex clear; branches opposite, decussate, horizontal and drooping; branchlets terete. *Bark* smooth or scaly, brown or dark brown; inner bark yellow. *Terminal bud* concealed between the bases of the uppermost pair of petioles. *Leaves* opposite, decussate; lamina oblong, oblong-obovate or obovate, 13–26 × 5–8.5 cm, apex acute, acuminate or mucronate, base cuneate, margin entire, coriaceous, shiny dark green above, paler below, glabrous on both surfaces, midrib grooved above and raised below, secondary veins 15–29 pairs, curving towards the margin connected in distinct loops and united into an intramarginal vein, with intersecondary veins, veinlets reticulate, visible on both surfaces, with a few scattered black gland dots on both surfaces, interrupted long wavy lines of differing lengths, nearly parallel to the midrib, running across the secondary veins to the margin; petiole 1.2–2.5 cm long, 2.5–4 mm in diam., grooved above, glabrous, with a small basal appendage clasping the branch; young leaves and petiole red; fresh leaves crispy when crushed; dry leaves chartaceous, blackish-brown, dark brown or greenish-brown. *Inflorescences* terminal, axis thick in male. *Flowers* unisexual, plants dioecious, 4-merous; bracts caducous; sepals and petals opposite, decussate, glabrous; sepals concave, reddish-green, turning reddish-pale yellow after falling off, the outer pair smaller than the inner pair; petals bright red. *Male flowers* in a thyrse, 7–10 cm long, 10–22-flowered, 4–5.5 cm in diam.; bracts lanceolate, 0.6–1 cm × 2–3 mm, apex acute; pedicel 2–2.5 cm long, 3–5 mm in diam.; sepals 4, suborbicular or orbicular, apex rounded, the outer pair 1–1.5 × 0.9–1.5 cm, the inner pair 1.6–2.3 × 1.7–2.5 cm; petals 4, suborbicular or broadly elliptic, 1.6–2.5 × 1.5–2.2 cm, apex rounded; stamens numerous, united in a ring central mass surrounding a pistillode, 1.1–1.4 cm in diam., 5.5–6 mm high; anthers sessile, 2-thecous (of 4 pollen sacs), ellipsoid, 0.5–1.5 × 0.5–1 mm, longitudinally dehiscent; pistillode fungiform (mushroom-shaped), 6–7.5 mm long, stigma bright red, sessile, convex, 0.7–1 cm in diam. *Female flowers* solitary, sometimes in a pair, 4–5.5 cm in diam.; bracts

lanceolate, 0.6–1.3 cm × 2–3 mm, apex acute; pedicel 1–2.5 cm long, 5–8 mm in diam.; sepals 4, suborbicular or orbicular, apex rounded, the outer pair 1.5–1.8 × 1.4–1.6 cm, the inner pair 1.7–2 × 1.7–2 cm; petals 4, broadly elliptic or broadly obovate, 1.9–2.7 × 1.3–2 cm, apex obtuse or rounded; staminodes numerous, united in a short ring surrounding the base of the ovary; pistil fungiform, 1–1.5 cm long; ovary pale green, subglobose, 1.2–1.7 cm in diam., shallowly 10–14-lobed, 10–14-locular; stigma bright red, sessile, convex, 1.5–2 cm in diam., radiate, 10–14-lobed, papillate. *Fruits* a berry, depressed globose, 5.7–7 × 7.8–10 cm, 10–14-lobed and sulcate, concave at both ends, green, turning bright yellow when ripe, glabrous, pericarp fleshy, 2.2–3 cm thick, cut fruits with sticky, pale yellow latex; persistent stigma brown, concave, 1.6–2 cm in diam., not lobed, slightly papillate; sepals persistent and becoming a little larger than at flowering, reddish-green; fruiting stalk 1.5–4 cm long, 7–7.5 mm in diam. *Seeds* 11–13, pale yellow, reniform (kidney-shaped), 1.2–1.4 cm × 6–8.5 mm, 3.8–5 mm thick, obtuse at both ends, with yellow sarcotesta (Figures 2 and 3).

*Recognition*. *Garcinia atroviridis* is a dioecious tree up to 20 m tall; branches horizontal and drooping; branchlets terete; inner bark with colourless latex; flowers large, 4–5.5 cm in diam.; petals bright red; stamens numerous, united in a ring central mass surrounding a pistillode; stigma bright red; fruits depressed globose, 5.7–7 × 7.8–10 cm, 10–14-lobed and sulcate, concave at both ends, green, turning bright yellow when ripe; leaves oblong, oblong-obovate or obovate, 13–26 × 5–8.5 cm, shiny dark green above, with a few scattered black gland dots on both surfaces, interrupted long wavy lines of differing lengths; young leaves red; dry leaves chartaceous, blackish-brown, dark brown or greenish-brown.

*Distribution*. India (Assam, Arunachal Pradesh), Myanmar, Thailand, Peninsular Malaysia (Kedah, Penang, Perak, Selangor, Malacca, Johor), Singapore, Indonesia (Sumatra, Borneo) (Figure 4).

*Additional Specimens Examined*. THAILAND. PENINSULAR. Trang [Khao Chong Botanical Garden, Chong Subdistr., Na Yong Distr., 70 m alt., tree 10 m tall, male fl. red (cultivated), 16 February 2022, *Ngernsaengsaruay et al. G34-16022022* (BKF dry and spirit collections, QBG); ibid., tree 14 m tall, female fl. red, y. fr. (cultivated), 8 March 2022, *Ngernsaengsaruay & Boonthasak G35-08032022* (BKF dry and spirit collections, QBG); ibid., tree 14 m tall, y. fr. (cultivated), 8 March 2022, *Ngernsaengsaruay & Boonthasak G36-08032022* (BKF dry and spirit collections, QBG)]; Pattani [Khao Kalakhiri, in evergreen forest, c. 400 m alt., tree c. 20 m high, male fl., 31 March 1928, *Kerr 14913* (BKF, BM, K); Sai Khao Waterfall, Khok Pho Distr., c. 50 m alt., sterile (probably cultivated), 13 November 1968, *Smitinand 10506* (BKF)]; Yala [Betong Distr. (originally "Pattani" on the label), in evergreen forest by stream, c. 400 m alt., tree c. 15 m high, fr., 2 August 1923, *Kerr 7477* (BM, K); Bang Lang National Park, Aiyoeweng Subdistr., Betong Distr., in tropical evergreen rain forest, 550 m alt., sapling 2.5 m tall, sterile, 27 May 2022, *Ngernsaengsaruay G37-27052022* (BKF)]; Narathiwat [To Mo, Sukhirin Distr. (originally "Pattani" on the label), c. 61 m (200 ft) alt., tree 15 m high, female fl., y. fr., 14 April 1931, *Lakshnakara 585* (BM, K); Khao Re Chaw, To Mo, Sukhirin Distr., c. 550 m (1800 ft) alt., tree, female fl. red, 20 April 1931, *Lakshnakara 734* (BM, K); Nikhom Waeng, Weang Distr., in evergreen forest, 200 m alt., tree 6 m high, male fl., 28 February 1974, *Larsen & S.S. Larsen 32729* (K, P [P05062480]); ibid., *Larsen & S.S. Larsen 32729* (AAU, P); Ban Bang Khun Thong, Tak Bai Distr., along the swamp margin, tree 15 m high, fr. yellow, 11 July 1983, *Niyomdham 660* (BKF); Sirindhorn Peat Swamp Forest, Su-ngai Kolok Distr., fr., 17 June 2003, *Ngernsaengsaruay 372* (BKF spirit collection); Peninsular Arboretum, Su-ngai Kolok Distr., small tree 8 m high, male fl. red (cultivated), 20 February 2005, *Niyomdham & Puudjaa 7224* (BKF); Su-ngai Kolok Distr., in evergreen rain forest, moist and slightly shaded places, 30 m alt., tree up to 10–20 m tall, female fl., fr. yellow, 2 February 2017, *Upho UBON 543* (QBG)]; [Locality not specified, fl., s.d., *Winit 56* (BKF)] (Figure 4).

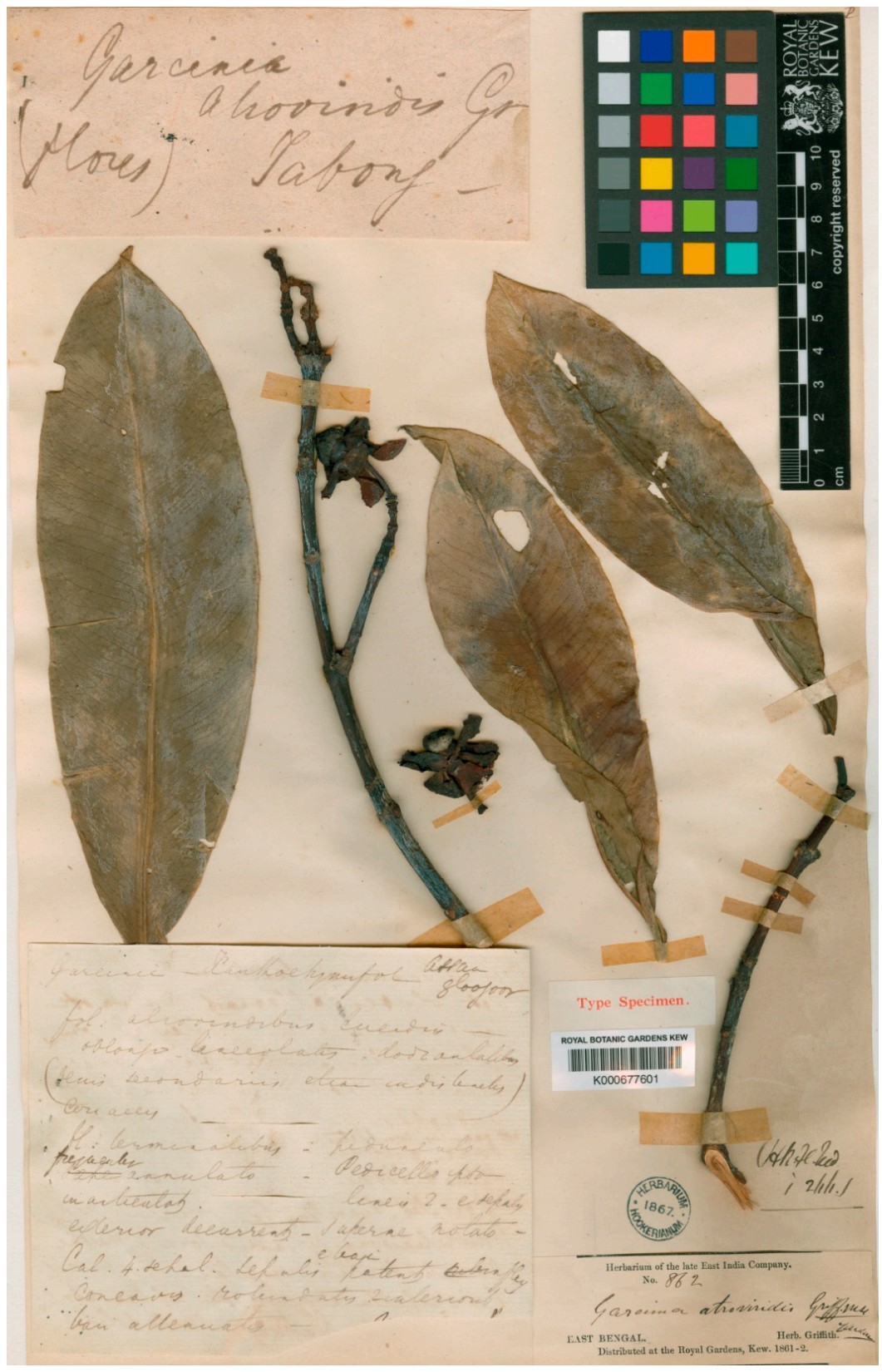

**Figure 1.** Lectotype of *Garcinia atroviridis*, *Griffith 862* (K [K000677601!]) from India, Upper Assam, Tabong, with female flowers and very young fruits (http://specimens.kew.org/herbarium/K0006776 01, accessed on 16 February 2022).

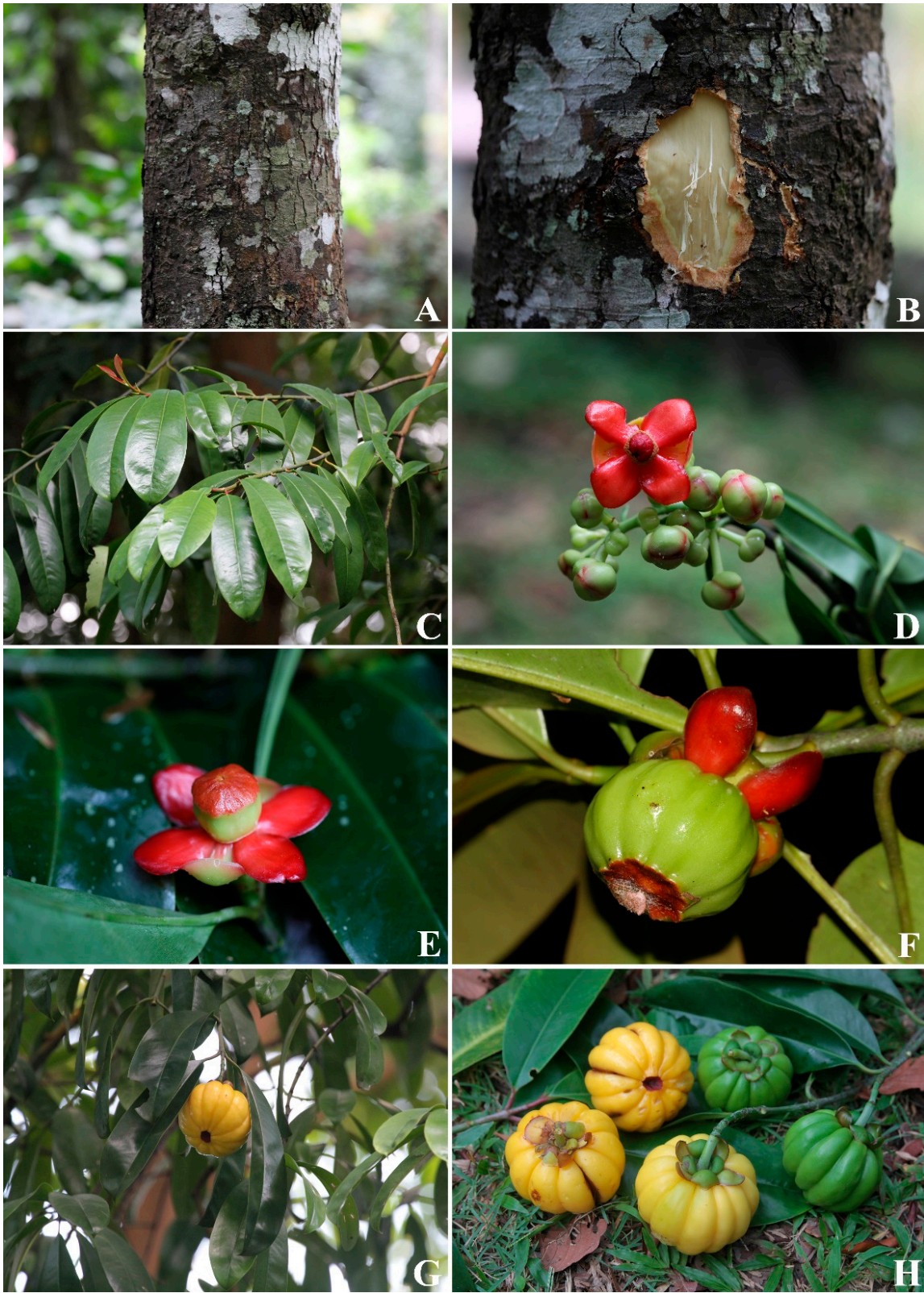

**Figure 2.** *Garcinia atroviridis*: (**A**) stem and outer bark; (**B**) outer bark and inner bark with colourless latex; (**C**) branches, young and mature leaves; (**D**) inflorescence with male flowers; (**E**) female flower; (**F**) young fruit; (**G**) fruiting branch; (**H**) mature and ripe fruits. Photos: Chatchai Ngernsaengsaruay.

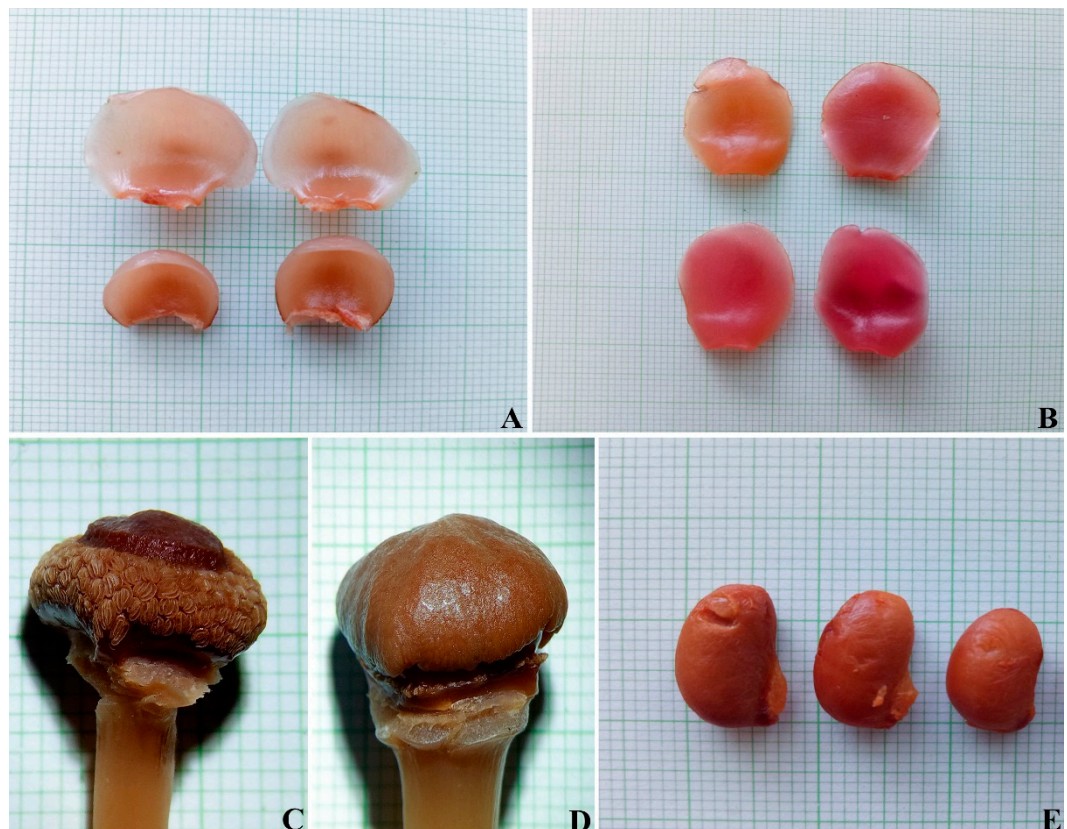

**Figure 3.** *Garcinia atroviridis*: (**A**) outer pair of sepals (lower) and inner pair of sepals (upper); (**B**) outer pair of petals (lower) and inner pair of petals (upper); (**C**) male flower showing stamens numerous, united in a ring central mass surrounding a pistillode (sepals and petals removed); (**D**) female flower showing pistil and staminodes (sepals and petals removed); (**E**) seeds. Photos: Weereesa Boonthasak.

MALAYSIA. Malacca, s.d., *Maingay* (Kew Distributed no. 154) (P [P05062479]); Perak, female fl., very y. fr., 1890, *Scortechini 627a* (G-DC [G00458945]; L [L2408805], P [P0506247 8]); Mersing, Johore, fr. green, turning yellow, 21 July 1957, *Abdullah 77858* (L [L2408773]); Perak, Kampong Dendang, Bruas, Dindings, female fl., fr., 31 October 1958, *Sinclair 9910* (L [L2408806]); Near road to Kedah Peak, Kedah, fr. green, ripe fr. yellow, 13 June 1966, *Kochummen FRI 2007* (L [L2408803]); Gunong Bubu, Perak, male fl., 15 February 1967, *Whitmore FRI 971* (L [L2408798, L2408799]); South of Kuala Lompat Krau Game Reserve, Central Pahang, fl., 16 April 1967, *Whitmore FRI 3571* (L [L2408796]); Piah F. R., Kuala Kangsar, Perak hillside, fr. yellow, 13 July 1967, *Kochummen FRI 2442* (L [L2408802]); Trengganu, Jambu Bongkok Forest Reserve, sterile, 20 December 1967, *T. & P. 11* (L [L2408775]); Ulu Muda, N. Kedah, fl. red, fr. green, 19 January 1969, *Bray FRI 11546* (L [L2408777]); Lowlands below G. Besar massif, 2 miles E of Kg., Tepoh Labis F. R. Johore, male fl., 18 March 1970, *Everett FRI 14095* (L [L2408800]); Semangkok F. R. North Selangor., male fl. red, 5 May 1970, *Chan FRI 13275* (L [L2408774]); Ulu Sungai Sepia near Kuala Aur, Pahang, fr. dark green, turning yellowish, 17 July 1970, *Shah & Noor 1945* (L [L2408776]); Hulu Perak, S. Perak nr., Fort Tapong, K. Kendrong, male fl., 26 January 1971, *Whitmore FRI 15799* (L [L2408810]) Perak, Lenggong Distr., Kampong Gelok, female fl., 13 March 1971, *Chin 950* (L [L2408781]); NW Pahang, Sg. Bertam at Kuala Mensun, Bamboo ridge top, male fl., 2 June 1971, *Whitmore FRI 20088* (L [L2408807]); NW Pahang Ulu Sg. Boh., fl., 3 June 1971, *Whitmore FRI 20114* (L [L2408809]); Trengganu mountains Sg Kerbat nr. Kuala Kerbat, Jeram Garok North of River, fr. green, 24 June 1971, *Whitmore FRI 20229* (L [L2408808]); Perak, Maxwells Hill, Larut Hills, Forest Reserve, female fl., fr. bright yellow, 25 August 1971, *Whitmore FRI 20389* (L [L2408804]); Kedah, Bukit Tam Mik, rubber estate land, N of Butter-

worth, male fl., 13 March 1980, *Stone 14486* (L [L2408797]); Kp. Sekam, NE of Tapah, Perak, male fl., fr. yellow, 5 September 1982, *Avé 127* (L [L2408801]); Pahang, Lesong F. R., Base Camp at Sg. Kindrin Riverine, fr. green, 18 July 1992, *Chua* et al. *FRI 39162* (L [L3878514]); Perak, Hulu Perak, Belum F. R., Sg. Sengum, female fl., fr. greenish to yellowish, 2 June 1998, *Kamarudin* et al. *FRI 42339* (AAU, BKF, L [L3812956]).

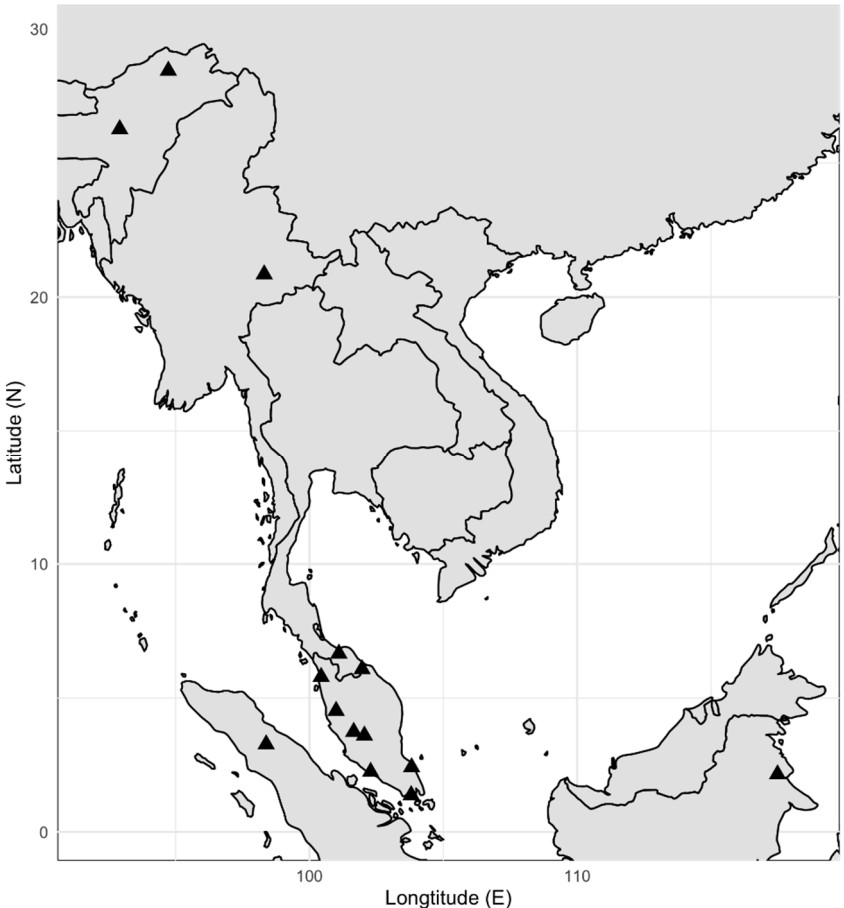

**Figure 4.** Distribution of *Garcinia atroviridis*.

SINGAPORE. Bukit Timah Nature Reserve, mature fr. green, ripe fr. yellow, 11 July 2019 (*Ngernsaengsaruay own observation*, with photos).

INDONESIA. Archipel. Ind., Sumatra, O. Kust [= East Coast], Sibolangit, 7 December 1928, *Karta 64* (L [L2408778]); E. Borneo, Berau, Tdg. Redeb, Kelai R., female fl., fr. green, 1 October 1963, *Kostermans 21023* (L [L2408780]); Sumatra, Sekundur Forest Reserve, E side of Gunung Leuser Natural Park, Langkat, Province of North Sumatra, Base camp at Aras Napal, upper Besitang River area, fr. green, 5 August 1991, *de Wilde*, *de Wilde-Duyfjes 21256* (L [L2408779]).

*Habitat and Ecology.* It is found in tropical lowland evergreen rain forests, peat swamp forests and along streams, near above mean sea level up to 550 m alt.

*IUCN Conservation Status. Garcinia atroviridis* is widely distributed from India to the Malay Peninsula and Indonesia, and has a large EOO of 1,177,468.29 km$^2$ and AOO of 192 km$^2$. In Thailand, this species is known from the natural distribution in the peninsular region (Pattani and Narathiwat Provinces), and has an EOO of 7120.08 km$^2$ and AOO of 48 km$^2$. Therefore, I consider the conservation assessment here as Least Concern (LC).

*Phenology.* Flowering January to May; fruiting February to August; flowering and fruiting more than once a year.

*Etymology*. The specific epithet *atroviridis* from Latin compound words, *atro-* meaning black or dark, and *viridis* meaning green, refers to the colour of the leaves, dark green [61–63].

*Vernacular Name*.    Cha muang chang (ชะมวงช้าง) (Peninsular), Ma kham khaek (มะขามแขก) (Narathiwat); **Som khaek (ส้มแขก)** (Trang, Pattani); Som khwai (ส้มควาย) (Trang); Som pha ngun (ส้มพะงุน) (Pattani); Som ma won (ส้มมะวน) (Peninsular); Som ma on (ส้มมะอน) (Pattani); A-sae-ka-lu-ko (อาแซกะลูโก) (Malay-Narathiwat); Asam gelugor, Asam gelugur (the fruit), Gelugor, Gelugur, Kayu gelugur (the tree) (Malaysia, Indonesia).

*Uses*. *Garcinia atroviridis* is commonly cultivated for its fruits in southern Thailand. The pericarp, sarcotesta, young shoots and leaves are edible and have a sour taste. The fruits are used fresh or dried, cooked or raw; the young shoots and leaves are used fresh and cooked, as a vegetable. The fruits are usually sliced, sun-dried and preserved for consumption in curries (southern Thai spicy sour yellow curries with fish: "Kaeng Som" (sour curry) or "Kaeng Lueang" (yellow sour curry)). It can be used as a pickle, tea and beverage flavouring, and it can be processed to make preserved fruit in syrup or it can be sun-dried. The fruits, young shoots and leaves are used as a sour flavouring in soups with pork, beef or fish. It can be used as a substitute for tamarinds or limes (the author's observations and interviews), in conformity with Agarwal [64], Bircher & Bircher [65], Burkill [66], Sastri [67], and Verheij & Coronel [50]. In Singapore, it is a tree suitable for gardens, parks and roadsides (the author's observation).

*Garcinia atroviridis* (Asam keping) is an important ingredient in most Malay dishes, where it is used as a seasoning or sour relish. In addition to the Asam keping, many value-added products, including juice, candy, chutney and tea, have been developed using the fresh fruits and leaves [68]. The fruits are too sour to be eaten raw but are tasty when stewed with sugar [66,67].

Medicinally, the fruits and leaves are applied to women after childbirth [50,66,68] and a decoction of leaves and roots is used in the treatment of earache [50,64,66,67]. The fruits are commonly used in diets in Southeast Asia [69]. The dried fruits are used as a fixative for dyes [50,64–67].

*Notes*. *Garcinia atroviridis* was named by Anderson, who cited two gatherings: *Griffith 862* from Upper Assam, Tabong and *Maingay s.n.* from Malacca [15]: 266. Following Art. 9.6 of the ICN [23], they constitute syntypes, in conformity with Jones [22]. Maheshwari cited the specimen "*Griffith*, Kew distrib. 862" in K as the type for the name *G. atroviridis* [7]. Therefore, this name has been lectotypified, following Art. 9.3 and 9.12 of the ICN [23]. I located the cited collection in K with barcode K000677601.

*Garcinia atroviridis* is a medium-sized tree, up to 27 m tall [6], but from my field observations and examined specimens, it grows up to 20 m tall, in conformity with Jones [22] and Kurz [42]. The inner bark of this species has a slightly transparent to clear yellow sap [6], but from my field observations, it has a little clear latex (colourless), in conformity with Corner [34]. The fruits of this species have 12–16 ribs and grooves [34], but I found 10–14-lobed and sulcate fruits in this study.

3.1.2. *Garcinia lanceifolia* Roxb. (Roxburgh, 1814, as G. lanceæfolia, nom. nud.; Roxburgh 1832)

*Garcinia lanceifolia* Roxb. [7]: 125, t. 3, fig. 21; [12]: 116; [15]: 263; [32]: 52; [39]: 106; [41]: 87; [42]: 91; [45]: t. 80, fig. D–E, t. 81, fig. A; [47]: 42; [48]: 623; [51]: 19, t. 118,119; [52]: 429; [54]: t. 103. Type: India, cultivated in H.B.C. (Calcutta Botanic Garden), female fl., y. fr., s.d., *Unknown collector*, *East India Company Herbarium 4861B* (lectotype K-W [K000639523!], isolectotypes CAL (not seen), P [P04700745!, P04700755!], designated here) (Figure 5).

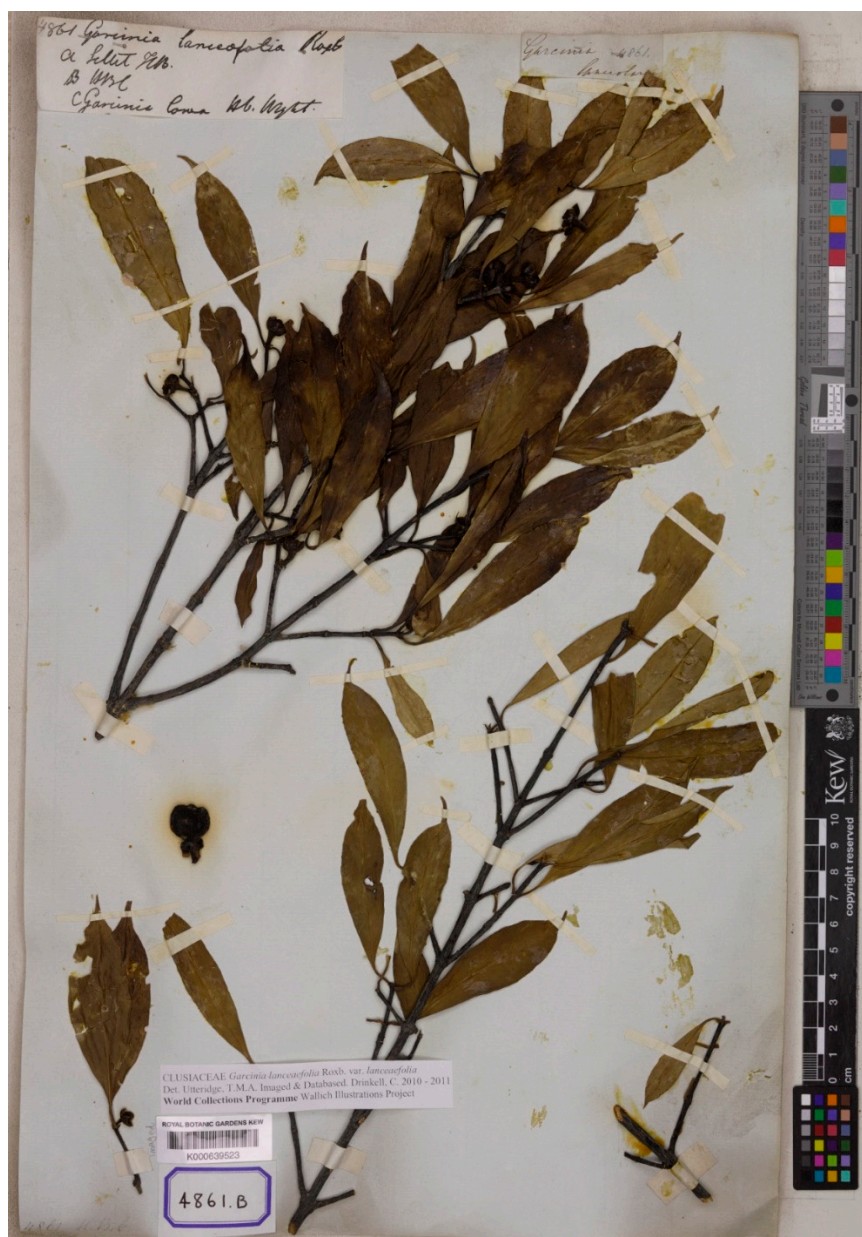

**Figure 5.** Lectotype of *Garcinia lanceifolia*, *Unknown collector*, *East India Company Herbarium 4861B* (K-W [K000639523!]) from India, cultivated in H.B.C. (Calcutta Botanical Garden), with female flowers and young fruits (http://specimens.kew.org/herbarium/K000639523, accessed on 16 February 2022).

*Garcinia gracilis* Pierre (Pierre, 1882) [14]: 260; [17]: 115; [19]: 353, fig. 543; [38]: 563, fig. 1557; [44]: t. 63; [46]: 301; [51]: 19. t. 142,143. Type: Cambodia, Ad flumen Se Kemoun, male fl., Jan. 1877, *Harmand 3618* (lectotype P [P04701635!], isolectotype P [P04701631!], designated here) (Figure 6).

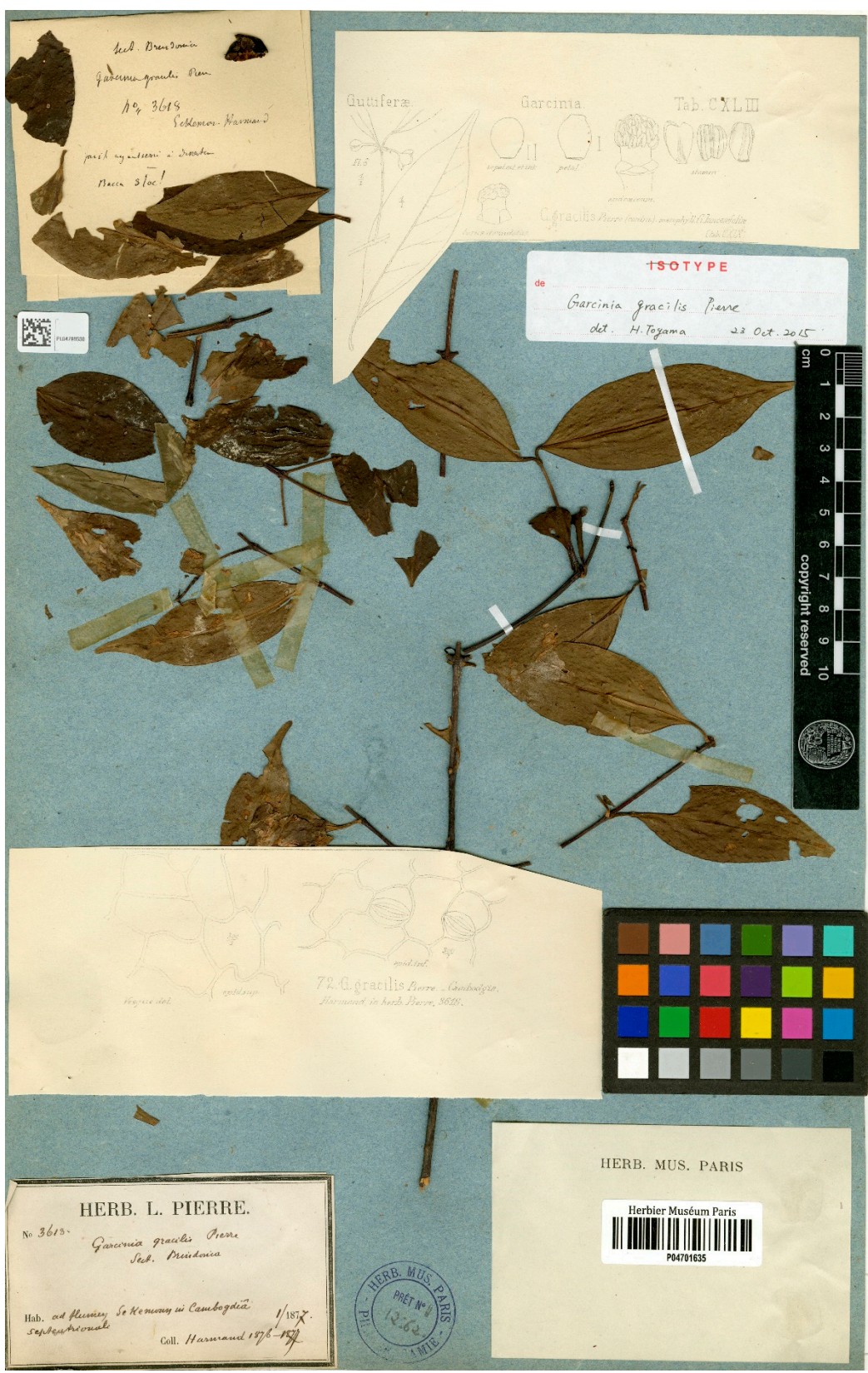

**Figure 6.** Lectotype of *Garcinia gracilis*, *Harmand 3618* (P [P04701635!]) from Cambodia, Ad flumen Se Kemoun, with male flowers (http://coldb.mnhn.fr/catalognumber/mnhn/p/p04701635, accessed on 16 February 2022).

*Lectotypifications*. Maheshwari cited *Wallich Cat. 4861A, 4861B* (CAL) from Sylhet as the type of *Garcinia lanceifolia* [7]. The type material must be from amongst the material cited in Flora Indica 1832. The material for the description must have been collected before 1815, when Roxburgh died, because even though it was only published in 1832, the R beside the name means that it is Roxburgh's name and description from his manuscript and not one of Wallich's later additions. It is certainly the case that any material collected by Wallich himself is very unlikely to be original material as he really only became active after Roxburgh had already died. I think this rules out the Bruce collection (H. Bruce is a plant collector for Wallich in Sylhet and Chittagong) in *4861A* but the H.B.C. (Calcutta Botanic Garden) collection in *4861B* could have been a Roxburgh specimen brought into the East India Company Herbarium. The problem may lie in citing the specimens as *Wallich 4861*. The specimen *4861B* is from H.B.C. collected by an unknown collector at an unknown time, but that it is likely to be original material. It is possible that Roxburgh described it from live plants and did not make a specimen, but I also think it is safe enough to assume this H.B.C. plant could have been from Roxburgh. I located three sheets of the specimen *4861B* collected from the same locality: one sheet at K-W [K000639523] and two sheets at P [P04700745, P04700755], and following Art. 9.6 of the ICN [23], they constitute syntypes. Among them, *Unknown collector, East India Company Herbarium 4861B* (K-W [K000639523]) is in the best condition and clearly shows the diagnostic characters for the species. It is selected here as the lectotype, following Art. 9.3 and 9.12 of the ICN [23].

*Garcinia gracilis* was named by Pierre based on the type *Harmand 3618* in Herb. L. Pierre from Cambodia [44]. I traced two sheets of the specimen *Harmand 3618* at P [P0470163 1, P04701635] collected from the same locality, and following Art. 9.6 of the ICN [23], they constitute syntypes. Therefore, the complete and well preserved specimen *Harmand 3618* at P [P04701635] is selected here as the lectotype, following Art. 9.3 and 9.12 of the ICN [23].

*Description*. Shrubs or small evergreen trees, 1–6 m tall, 12–25 cm girth; latex yellow, sticky; branches opposite, decussate, horizontal; branchlets 4-angular, very young branchlets red. *Bark* smooth or longitudinally very shallowly fissured, greyish-brown; inner bark reddish-brown. *Terminal bud* concealed between the bases of the uppermost pair of petioles. *Leaves* opposite, decussate; lamina elliptic, narrowly elliptic, oblong-elliptic or ovate, 4.5–10 × 1.5–3 cm, apex caudate or acuminate, base cuneate, margin repand or entire, subcoriaceous, shiny green to dark green above, paler below, glabrous and with scattered black gland dots on both surfaces, midrib shallowly grooved above, raised below, secondary veins 4–8 pairs, faint above, invisible below, but visible on both surfaces in dry leaves, curving towards the margin connected in distinct loops and united into an intramarginal vein, flattened above, slightly raised below, with intersecondary veins, veinlets reticulate, visible on both surfaces; petiole 0.4–1 cm long, 1–1.5 mm in diam., grooved above, twisted, glabrous, with a small basal appendage clasping the branch; fresh leaves crispy when crushed; young leaves shiny pale green and petiole red or greenish-red. *Inflorescences* terminal or axillary, cymose, in fascicles of 2–3 flowers or solitary. *Flowers* unisexual, plants dioecious, sometimes polygamo-dioecious, 4-merous, fully opened flowers with a small apical opening; bracts caducous, lanceolate or triangular, 1.2–7 × 1–2.3 mm, apex acute or obtuse; pedicel colour same as sepals and petals, thick, glabrous; sepals and petals opposite, decussate, concave, green or orangish-green in young flower buds, turning orangish-red or red, glabrous; sepals fleshy, the outer pair slightly larger than the inner pair; petals thinner than sepals, subequal. *Male flowers* 5–8 mm in diam.; pedicel 3.5–7 mm long, 2–3.5 mm in diam.; sepals 4, the outer pair suborbicular or orbicular, 4.5–6.5 × 5–6 mm, apex rounded, the inner pair suborbicular or broadly elliptic, 4–6 × 4–5.7 mm, apex rounded; petals 4, suborbicular or orbicular, 4–5 × 3.5–4.5 mm, apex rounded; stamens 28–36, united in a central short column; filaments very short; anthers 2-thecous, ellipsoid or broadly ellipsoid, 0.6–1 mm long, longitudinally dehiscent; pistillode absent. *Female flowers* 7–8.5 mm in diam.; pedicel 3–7 mm long, 5.5–4.5 mm in diam.; sepals 4, the outer pair suborbicular or orbicular, 5–6.5 × 5–6 mm, apex rounded, the inner pair suborbicular or broadly elliptic, 5–6 × 4–5.8 mm, apex rounded; petals 4, suborbicular or orbicular, 3–5

× 3–4.5 mm, apex rounded; staminodes 10–20; filaments 2–2.5 mm long, basally united in several bundles surrounding the base of the ovary but distally free; anthers ellipsoid or broadly ellipsoid, 0.8–1 × 0.4–0.6 mm; pistil fungiform, 3–4 mm long; ovary pale green, globose or subglobose, 4–4.5 mm in diam., shallowly lobed or indistinctly lobed, 5–7-locular; stigma pale yellow, sessile, convex, 2.5–3 mm in diam., radiate, shallowly 5–7-lobed, papillate. *Fruits* a berry, depressed globose or subglobose, 2.2–3.5 × 2.3–4.5 cm, very shallowly 5–7-lobed or indistinctly lobed, apex usually concave, green, orangish-yellow, turning orange or bright red when ripe, glabrous, glossy, pericarp fleshy, 5–7 mm thick, cut fruits with sticky, yellow latex; persistent stigma dark brown, flattened, 2–3 mm in diam., radiate, shallowly 5–7-lobed, papillate; sepals persistent and becoming larger than at flowering, greenish-red or red; fruiting stalk 0.5–1 cm long, 3–4 mm in diam., colour same as persistent sepals. *Seeds* 1–7, brown, compressed, reniform, 0.8–1.5 × 0.5–1 cm, obtuse at both ends, with white sarcotesta (Figure 7).

*Recognition. Garcinia lanceifolia* is characterised as a dioecious, sometimes polygamo-dioecious, shrub or small tree, 1–6 m tall, with yellow latex; branchlets 4-angular; inflorescences in fascicles of 2–3 flowers or solitary; flowers small, 5–8.5 mm in diam., fully opened flowers with a small apical opening (look like flower buds); sepals and petals orangish-red or red; stamens many, united in a central short column; fruits depressed globose or subglobose, 2.2–3.5 × 2.3–4.5 cm, very shallowly 5–7-lobed or indistinctly lobed, apex usually concave, green, orangish-yellow, turning orange or bright red when ripe, glossy; leaves elliptic, narrowly elliptic, oblong-elliptic or ovate, 4.5–10 × 1.5–3 cm, shiny; young leaves shiny pale green and petiole red or greenish-red.

*Distribution.* India (Assam), Bangladesh (Chittagong hills), Myanmar, Vietnam, Laos, Cambodia, Thailand (Figure 8).

*Additional Specimens Examined.* THAILAND. NORTH-EASTERN. Bueng Kan [Mueang Bueng Kan Distr. (originally "Chaiyaburi, Nong Khai" on the label), in evergreen forest, tree c. 4 m high, fr., 22 February 1924 (as *G. gracilis*), *Kerr 8540* (BM, C, E [E00839777], K, L [L2403602], P [P04701630]); Phon Charoen Distr., shrub 3 m high, fr., 23 June 1997 (as *G. gracilis*), *Niyomdham 5114* (BKF); Phu Wua Wildlife Sanctuary, Bung Khla Distr., nature trail to Lat Plueai Waterfall, in dry evergreen forest, 200 m alt., shrub c. 1 m tall, fr., 21 May 2004 (as *G. gracilis*), *Pooma* et al. *4169* (BKF); Phu Wua Wildlife Sanctuary, Bung Khla Distr., shaded in lowland dry evergreen dipterocarp forest, 200 m alt., shrub 1–2 m tall, fr., 3 May 2002 (as *G. gracilis*), *Pooma* et al. *3443* (BKF); Mueang Bueng Kan Distr., in open areas, 170 m alt., shrub 2 m high, fl., 27 December 2011 (as *Garcinia* sp.), *Norsaengsri & Tathana 8601* (BKF, QBG)]; SOUTH-WESTERN: Kanchanaburi [Ban Sane Phong, Sangkhla Buri Distr., along stream in mixed evergreen/deciduous forest, 220 m alt., female fl., December 2003 (as *G. merguensis*), *Kansuntisukmongkol 1040* (CMUB); Thong Pha Phum National Park, Thong Pha Phum Distr., small tree 3 m tall, fr., 24 April 2004, *Ngernsaengsaruay G16-24042004* (BKF spirit collection); SOUTH-EASTERN: Chachoengsao [Khao Aang Rue Nai, shrub 3–5 m high, fr., 6 May 1997 (as *Garcinia* sp.), Niyomdham 5014 (AAU, BKF); Lum Chang Wat Wildlife Reserve, about 1 km from headquarters, in disturbed lowland forest, 100 m alt., tree to 2 m tall, fr., 16 February 2004 (as *G. parvifolia*), *Wilkie* et al. *425* (BKF, E [E00180827]); Lum Chang Wat Forest Protection Unit, Khao Aang Rue Nai Wildlife Sanctuary, fl., 7 January 2008 (as *G gracilis*), *Phonsena & Chusithong 5793* (BKF); Khao Hin Son Botanic Gardens, Khao Hin Son Subdistr., Phanom Sarakham Distr., small tree 2.5 m tall, fr. (cultivated), 23 February 2019, *Ngernsaengsaruay & Boothasak G17-23022019* (BKF dry and spirit collections, QBG); ibid., small tree 3 m tall, male fl., 23 February 2019, *Ngernsaengsaruay & Boothasak G18-23022019* (BKF dry and spirit collections, QBG); ibid., small tree 2 m tall, fr., 23 February 2019, *Ngernsaengsaruay & Boothasak G19-23022019* (BKF spirit collections); ibid., small tree 3.5 m tall, male fl., 28 January 2022, *Ngernsaengsaruay & Boothasak G20-28012022, G21-28012022* (BKF dry and spirit collections, QBG); ibid., small tree 3.5 m tall, female fl., 28 January 2022, *Ngernsaengsaruay & Boothasak G22-28012022, G23-28012022* (BKF dry and spirit collections, QBG)]; Rayong [Locality not specified, fr., February 1920 (as *G. gracilis*), *Collins 538* (E [E00839776], K, P [P04701629); Ban Phe, fr., 23 February 1930 (as *Garcinia* sp.) (BM),

(as *G.* cf. *merguensis*) (K)], *Put 2748* (BM, K); Ban Nong Hong, Kachet Subdistr., Mueang Rayong Distr., shrub, fr., 22 March 2005 (as *G. gracilis*), *Kertsawang 477* (QBG); Khao Sum Pratu, Kachet Subdistr., Mueang Rayong Distr., shrub, fr., 24 April 2006 (as *G. gracilis*), *Kertsawang 593* (QBG)]; Chanthaburi [Pong Nam Ron Distr., in evergreen forest by stream, 200 m alt., shrub 2 m tall, fl., 15 January 1958, *Smitinand 4662* (BKF); Khao Soi Dao, in dry evergreen forest, fl., 15 January 1958 (as *Garcinia* sp.), *Sørensen* et al. *368* (C); Khao Soi Dao Nuea, in disturbed evergreen forest, 300 m alt., tree 6 m tall, fr., 13 May 1974 (as *Garcinia* sp.), *Geesink* et al. *6766* (AAU, BKF, K, L [L2409521], P [P05061694]); Khao Soi Dao Wildlife Sanctuary, Soi Dao Distr., in mixed deciduous forest by stream, c. 200 m alt., shrub 3 m high, fr., 14 May 1995 (as *Garcinia* sp.), *Santisuk s.n.* (BKF100173); Khlong Saba, Khao Sip Ha Chan, Kaeng Hang Maeo Distr., in dry evergreen forest, small tree or shrub 2.5 m high, female fl., fr., 11 February 2007 (as *G. gracilis*), *Watthana 2210* (QBG); trail from Khlong Tani–Khlong Thap Mak, Khao Soi Dao Wildlife Sanctuary, Pong Nam Ron Distr., in dry evergreen forest, 700 m alt., small tree 2 m high, fl., 28 January 2008 (as *G. gracilis*), *Suksathan 4488* (QBG); Thap Sai Subdistr., Pong Nam Ron Distr., in evergreen forest, small tree 2 m high, fr., 18 March 2012 (as *G. gracilis*), *Sawangsawat 590* (QBG); Khao Soi Dao Wildlife Sanctuary, in evergreen forest near 16th waterfall, 600 m alt., sterile, 24 May 2013, *Tagane* et al. *T1651* (BKF)]; Trat [Khao Saming Distr., in evergreen forest, below 50 m alt., small tree c. 5 m high, fl., 1 January 1930 (as *Garcinia* sp.), *Kerr 17899* (BM, K); Ao Kong Kang and Ao Salak khok (originally "Aw Ong Kang and Salak Koh" on the label), Ko Chang, in evergreen forest on granitic hills, 50 m alt., shrub, fr., 8 May 1974 (as *Garcinia* sp.), *Geesink* et al. *6622* (L [L2409520]); Ko Kut, 0–50 m alt., fl., fr., 6 April 2002 (as *G.* cf. *atroviridis*), *Phengklai* et al. *14603* (BKF143190), the specimen *Phengklai* et al. *14603* and all data the same as BKF142164, but different species (BKF142164 = *G. cowa*)]; PENINSULAR: Ranong [Khlong Kam Phuan, disturbed evergreen forest along trail, treelet, fr., 26 April 1973 (as *Garcinia* sp.), *Geesink & Santisuk 4942* (BKF, AAU, C, L [L2409563]); Khlong Nakha Wildlife Sanctuary, Kamphuan Subdistr., Suk Samran Distr., 80 m alt., treelet 4 m tall, fl., 17 January 2005 (as *Garcinia* sp.), *Gardner & Tippayasri ST1343* (K); Ban Nai Wong, La-un Distr., beside stream on limestone, tree 3 m tall, fl., 19 February 2006 (as *Garcinia* sp.), *Middleton* et al. *3836* (BKF, E [E00261366])]; Surat Thani [Tha Chang Distr., in evergreen forest, small tree 2–3 m high, fr., 5 March 1974 (as *Garcinia* sp.), *Sangkachand 19* (AAU, BKF, C, E [E00839779], K, L [L2409477], P [P05062022]); Khlong Phanom National Park, Phanom Distr., park headquarters 'Big Tree' nature trail, edge of partially/disturbed evergreen forest, on rugged limestone terrain, 150 m alt., small tree 6 m high, fr. bright red, shiny, 16 June 2004 (as *Garcinia* sp.), *Setsin ST0774* (K); ibid., understorey of relatively undisturbed evergreen forest on rugged limestone terrain, 170 m alt., treelet 2 m tall, fr., 19 March 2005 (as *Garcinia* sp.), *Sidisunthorn & Tippayasri ST1695* (K); Khlong Phanom National Park, Phanom Distr., trail near park headquarters, in evergreen forest, 100 m alt., shrub 2 m tall, female fl., fr., 16 February 2005 (as *Garcinia* sp.), *Williams & Pooma 1543* (BKF, E [E00351572])]; Phangnga [Nop Pring, in scrub, c. 100 m alt., shrub c. 2.5 m high, fr., 6 March 1930 (as *Garcinia* sp.), *Kerr 18392* (BM, C, K, P [P05062033]); Ko Ra, Khura Buri Distr., in evergreen forest, shrub 3 m high, male fl., 28 January 2009 (as *G. merguensis*), *Watthana 2937* (BKF)] (Figure 8).

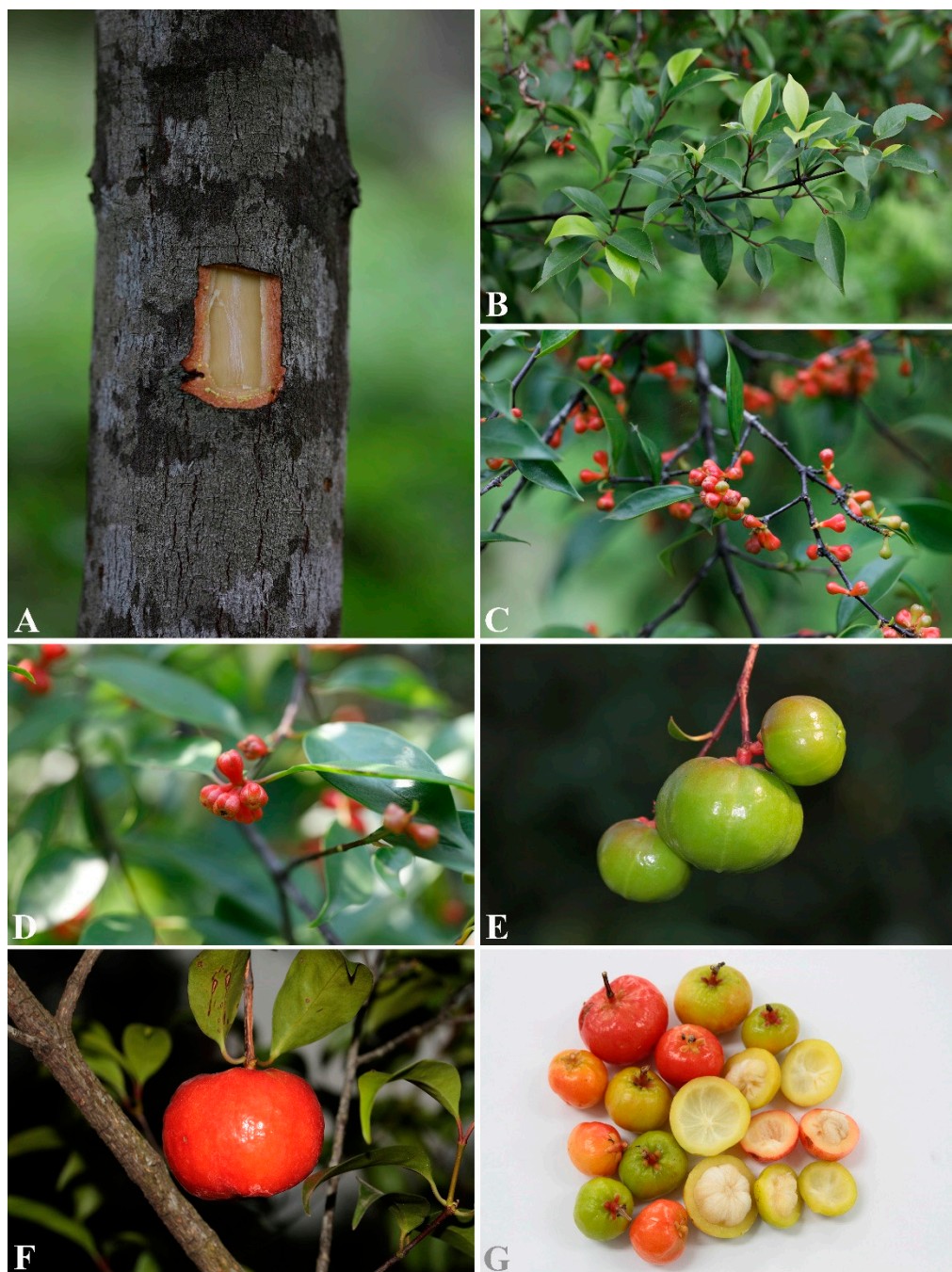

**Figure 7.** *Garcinia lanceifolia*: (**A**) stem, outer bark and inner bark with yellow latex; (**B**) branches, young and mature leaves; (**C**) male flowering branches; (**D**) inflorescence with female flowers; (**E**) fruiting branch; (**F**) ripe fruit; (**G**) mature and ripe fruits, showing pericarps and seeds with white sarcotesta (cross section). Photos: Chatchai Ngernsaengsaruay.

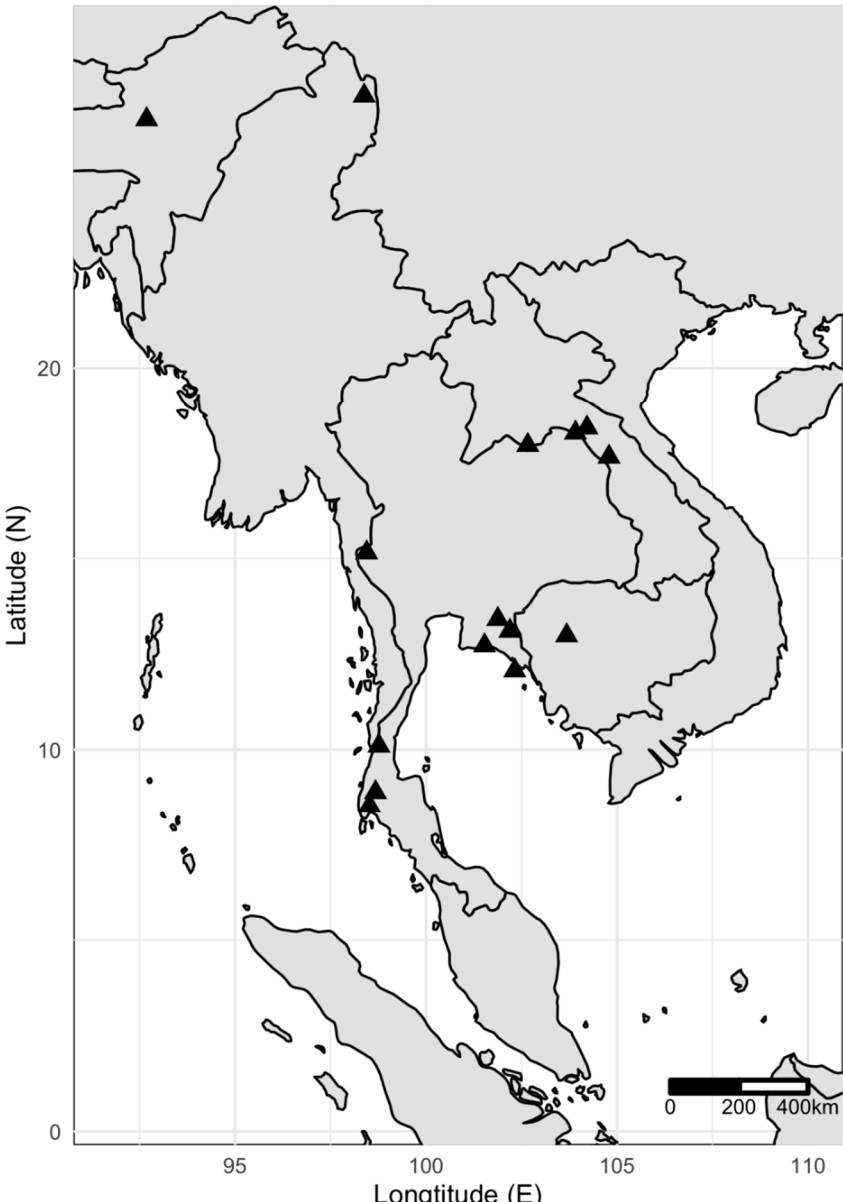

**Figure 8.** Distribution of *Garcinia lanceifolia*.

INDIA. *Roxburgh s.n.*, 1813, (BM [BM000611614]); Assam, Seppur, May 1908, *Alleizette s.n.* (L [L2403945]).

MYANMAR. Kachin State, near Mansi, Bhamo Distr., fr., May 1909, *Cubitt s.n.* (E [E00839500]); Sagaing, Katha Distr., 25 February 1914, *English 58* (E [E00839501]).

LAOS. Bassin d'Attapeu, Expedition 1875–1877, *Harmand s.n.* (P [P04701627]); Vientiane, 3 February 1952, *Vidal 1420* (P [P04701633]); Khammouane Province, near Talaki hill, Namahang Village, fl., 15 February 2002, *Soejartoet* et al. *11916* (L [L2409458, L2409489, L2409499]); Khammouane Province, Nakai Distr., noted that fr., 20 May 2006, *Nanthavong* et al. *BT451* (L [L2409493]); Khammouane Province, Nakai Distr., fr., 26 February 2007, *Vannachak* et al. *BT858* (E [E00702525], (L [L2409462]); Bolikhamxai Province, Ban Lak Xao, Khamkeut Distr., fr., 6 April 2005, *Svengsuksa* et al. *BT104* (L [L2409449, L2409450, L2409470]).

CAMBODIA. Dist: in sylvis ad lai long in prov. Lay neah gallear austro, January 1877, *Harmand 4014* (K [K000677691].

*Habitat and Ecology*. It is found in dry evergreen forests, tropical evergreen rain forests, mixed deciduous forests, along streams in dry evergreen and mixed deciduous forests and limestone, near above mean sea level up to 300(–700) m alt.

*IUCN Conservation Status*. *Garcinia lanceifolia* is widely distributed from India to Indo-China. It is known in many localities and has a large EOO of 3,157,674.46 km$^2$ and a relatively large AOO of 140 km$^2$. In Thailand, this species is known from the north-eastern, the south-western, the south-eastern and the peninsular regions, and has an EOO of 368,145.62 km$^2$ and an AOO of 84 km$^2$. This wide distribution and the number of localities from which it is known make it appropriate to consider its status as LC.

*Phenology*. Flowering December to April, with a peak in January to February; fruiting January to June, with a peak in March to May.

*Etymology*. The specific epithet *lanceifolia* from Latin compound words, *lancea-* meaning lance or spear, and *-folia* meaning -leaved, refers to the leaf shape, lanceolate (spear-shaped). The specific epithet of its synonym, *G. gracilis* is a Latin word meaning thin or slender, referring to the character of the leaves [61–63].

*Vernacular Name*. **Ma paem** (มะแปม), Mak paem (หมักแปม), Mak paem (หมากแปม) (Bueng Kan, Nong Khai, Laos); Cha maeng (ชะแมง) (Rayong); Cha mang (ชะมาง) (South-Eastern); Ma dan daeng (มะดันแดง) (Central); Salit (สลิด) (Tha Chang, Surat Thani); Assamikau, Kan tekera, Prango-arong, Prangsu, Rupohi-thekera (India-Assamese); Dieng-soh-jadu (India-Khasi); Pelte (India-Lushai); Thisuru (India-Garo).

*Uses*. *Garcinia lanceifolia* is often cultivated for its fruits in south-eastern Thailand. The pericarp, sarcotesta, young shoots and leaves are edible and have a sour taste. The ripe fruits are edible and are used fresh in beverages and jams and in flavoured ice cream. It can be processed to make preserved fruit in syrup, or it can be sun-dried. It is often cultivated as an ornamental plant (the author's observations and interviews). In India, it is often cultivated in villages for its fruits, which are acidic and eaten with relish. The leaves are cooked as a vegetable [64,67].

*Note*. The specimen *Phengklai* et al. *14603* (BKF143190) from Ko Kut, Trat Province has collector no. and all data are the same as *Phengklai* et al. *14603* (BKF142164). I located both specimens found as different species and identified the specimen *Phengklai* et al. *14603* (BKF143190) as *Garcinia lanceifolia* and *Phengklai* et al. *14603* (BKF142164) as *G. cowa*.

3.1.3. *Garcinia pedunculata* Roxb. ex Buch.-Ham. (Roxburgh, 1814, nom. nud.; ex Buchanan- Hamilton, 1827)

*Garcinia pedunculata* Roxb. ex Buch.-Ham. [7]: 119, t. 2, fig. 11; [12]: 121; [15]: 264; [16]: 43; [18]: 52, fig. 58; [19]: 361, fig. 551; [32]: 50; [33]: 45, t. 1; [37]: 351, figs. 1,2; [38]: 560, fig. 1547; [39]: 107; [45]: t. 79, fig. M; [47]: 42; [48]: 625; [51]: 20, t. 121; [52]: 374; [54]: t. 114,115. Type: India, Goalpara, female fl., 10 Oct. 1808, *Buchanan-Hamilton collection*, *East India Company Herbarium 4860A* (lectotype CAL, isolectotype K-W [K001104082!], designated by Maheshwari [7] (Figure 9).

*Garcinia planchonii* Pierre (Pierre, 1882) [14]: 261; [38]: 560, fig. 1548; [44]: t. 61; [46]: 306; [51]: 20, t. 153; [52]: 373, as *G. planchoni*. Type: Vietnam, Crescit ad ripas fluvii Dougnai juxta Chiao Xhan austro Cochinchinae, female fl., fr., Mar. 1873 (as *G. planchoni*), *Pierre 1313* (lectotype P [P04701313!], isolectotypes K [K000677692!, K000742483!], L [L0700333!], P [P04701308!, P04701309!, P04701312!, P04701314!, P04701315!, P04701316!, P04701319!], U [U1208239!], designated here). **syn. nov.** (Figure 10).

*Lectotypification*. *Garcinia planchonii* was named by Pierre based on the type *Pierre 1313* from Vietnam [44]. I located twelve sheets of this specimen collected from the same locality: two sheets at K [K000677692, K000742483], one sheet at L [L0700333], eight sheets at P [P04701308, P04701309, P04701312, P04701313, P04701314, P04701315, P04701316, P04701 319], and one sheet at U [U1208239], and following Art. 9.6 of the ICN [23], they constitute syntypes. Hence, the complete and well preserved specimen *Pierre 1313* at P [P04701313] is selected here as the lectotype, following Art. 9.3 and 9.12 of the ICN [23].

*Description*. Evergreen trees, up to 25 m tall, 70–160 cm girth, narrowly buttressed up to 1.8 m tall, usually found in large trees; latex clear; branches opposite, decussate, horizontal; branchlets weakly 4-angular. *Bark* scaly, brown; inner bark red. *Terminal bud* concealed between the bases of the uppermost pair of petioles. *Leaves* opposite, decussate; lamina obovate, sometimes obovate-oblong or elliptic, 18–31.5 × 7.5–12.5 cm, apex mucronate, obtuse or emarginate, base cuneate, margin repand, coriaceous, dark green above, paler below, glabrous and with a few scattered black gland dots on both surfaces, midrib grooved above and raised below, secondary veins 14–20 pairs, curving towards the margin connected in distinct loops and united into an intramarginal vein, with intersecondary veins, veinlets scalariform-reticulate, visible on both surfaces; petiole 1.5–3.8 cm long, 3.5–5.5 mm in diam., grooved above, slightly twisted, glabrous, with a small basal appendage clasping the branch; young leaves pale green; fresh leaves crispy when crushed; leaves turning yellow before falling off. *Inflorescences* terminal, axis thick in male. *Flowers* unisexual, plants dioecious, 4-merous, fully opened flowers with an apical opening; bracts 2, pale green or yellowish-green, caducous; pedicel thick, 4-sided, widened towards the upper part, pale green or yellowish-green, glabrous; sepals and petals opposite, decussate, pale green or yellowish-green, glabrous; sepals concave, fleshy, turning brownish-green after falling off, the outer pair slightly larger than the inner pair; petals slightly concave, thinner than sepals, subequal. *Male flowers* in a thyrse, 7–16 cm long, many-flowered, 0.8–1.2 cm in diam.; bracts broadly ovate or suborbicular, c. 3 × c. 3 mm, apex obtuse; pedicel 1.7–2.5 cm long, 5–7 mm in diam.; sepals 4, the outer pair suborbicular, 6.5–8.5 × 7.5–9.5 mm, apex rounded, the inner pair suborbicular or orbicular, 6.5–8 × 7–7.5 mm, apex rounded or obtuse; petals 4, ovate, 5.5–8 × 4–6 mm, apex obtuse; stamens 58–65, united in a central 4-sided column surrounding a pistillode, 5–7 mm long, filaments very short; anthers 2-thecous, subglobose or broadly ellipsoid, 0.6–1 × 0.8–1 mm, longitudinally dehiscent; pistillode columnar-cuneate, slightly angular. *Female flowers* solitary or in a cyme, c. 5 cm long, 2–5-flowered, 1.3–1.7 cm in diam.; bracts broadly ovate or suborbicular, c. 5 × 4–5 mm, apex obtuse; pedicel 1.3–4 cm long, 0.7–1.2 cm in diam.; sepals 4, the outer pair suborbicular, 1–1.3 × 1.1–1.5 cm, apex rounded, the inner pair suborbicular or orbicular, 0.9–1.2 × 0.8–1.3 cm, apex rounded; petals 4, ovate, 0.7–1 × 0.6–0.9 cm, apex obtuse; staminodes 36–55; filaments 0.3–1 mm long, basally united in several bundles surrounding the base of the ovary but distally free; anthers subglobose or broadly ellipsoid, 0.9–1.5 × 0.8–1 mm; pistil fungiform, 0.7–1 cm long; ovary pale green, subglobose, 6–8 mm in diam., shallowly 13–15-lobed, 7–10-locular; stigma yellow, sessile, convex, 5.5–7.5 mm in diam., radiate, 7–10-lobed, papillate. *Fruits* a berry, subglobose or slightly depressed subglobose, 6–10 × 7–12 cm, shallowly 13–15-lobed and sulcate, concave at both ends, green, turning bright yellow when ripe, glabrous, pericarp fleshy, 2.2–3 cm thick, inner layer of cut fruits with sticky, pale yellow latex; persistent stigma pale brown, concave, 0.6–1 cm in diam., radiate, shallowly 7–10-lobed, papillate; sepals persistent and becoming a little larger than at flowering, green, turning brownish-green; fruiting stalk 1.7–4 cm long, 1–1.5 cm in diam. *Seeds* 5–8, pale brown, ellipsoid, 2.4–2.8 × 1.6–1.8 cm, 0.8–1.1 cm thick, obtuse at both ends, with yellow sarcotesta (Figures 11 and 12).

*Recognition*. *Garcinia pedunculata* is characterised by its dioecious trees up to 25 m tall; branchlets weakly 4-angular; inner bark with colourless latex; leaves obovate, sometimes obovate-oblong or elliptic, 18–31.5 × 7.5–12.5 cm, veinlets scalariform-reticulate; fruits large, subglobose or slightly depressed subglobose, 6–10 × 7–12 cm, shallowly 13–15-lobed and sulcate, concave at both ends, green, turning bright yellow when ripe; flowers 0.8–1.7 cm in diam., fully opened flowers with an apical opening; pedicel thick, 4-sided, up to 4 cm long, widened towards the upper part; sepals and petals pale green or yellowish-green, turning brownish-green after falling off; male flowers in a thyrse of many-flowered; female flowers solitary or in a cyme of 2–5-flowered; stamens numerous, united in a central 4-sided column surrounding a pistillode.

*Distribution*. India (Assam), Bangladesh (Sylhet), Myanmar, China (Yunnan, Tibet), Vietnam, Laos, Thailand (Figure 13).

*Additional Specimens Examined*. THAILAND. NORTHERN. Mae Hong Son [Ban Huai Hom, Huai Hom Subdistr., Mae La Noi Distr., in semi-evergreen forest, 1200 m alt., female fl., fr., 1 October 2003 (as *Garcinia* sp.), *Pongamornkul 2198* (BKF, QBG); Ban Kai Dam, Mae La Noi Distr., in semi-evergreen forest, 900 m alt., female fl., 29 November 2003 (as *Garcinia* sp.), *Pongamornkul 2199* (QBG)]; Chiang Mai [Queen Sirikit Botanical Garden, Mae Ram Subdistr., Mae Rim Distr., 700 m alt. (cultivated), 2 August 2014 (as *Garcinia* sp.), *Titthi-wigrom 3* (CMUB)]; Chiang Rai [Doi Tung Royal Villa, 970 m alt., fr., 21 June 2002 (*Garcinia* sp.), *Chamchumroon* et al. *1540* (BKF); ibid. fr., 21 June 2002, *Ngernsaengsaruay & Chamchum-roon 78* (BKF spirit collection); Dong Pa Lan forest at the edge of Nong Luang reservoir, Wiang Chai Distr., in degraded deciduous hardwood forest with bamboo, 340 m alt., male fl., 2 March 2012 (as *Garcinia* sp.), *van de Bult 1303* (BKF, CMUB); Ban Pha Mi, Mae Sai Distr., by stream bank, 513 m alt., male fl., 30 March 2012, *Norsaengsri & Tathana 9343* (QBG); Pha Khong Cave, Phan Distr., in dry evergreen forest, 460 m alt., fr., 30 May 2016, *Muangyen 997* (QBG)]; Phayao [Doi Luang National Park, Champa Thong Waterfall, mostly shaded area, seasonal, mixed evergreen/deciduous, primary hardwood forest, along the stream, 600 m alt., fr., 7 May 1997, (as *Garcinia* sp.), *Maxwell 97-512* (CMUB); Champa Thong Waterfall, Mueang Distr., deciduous forest, 550 m alt., fr., 7 April 1999 (as *Garcinia* sp.), *Srisanga & Watthana 612* (QBG)]; Lamphun [Mae Ping, 130 m alt. (originally "Lampang" on the label), fr., 6 April 1930 (as *G. costata*), *Winit 1977* (BKF, K)]; Lampang [Chae Son National Park, above Don Chai Village off the dirt road to Pa Miang Village, Mueang Pan Distr., open disturbed, seasonal mixed evergreen/deciduous hardwood forest with much bamboo, 700 m alt., male fl., 16 February 1996 (as *Garcinia* sp.), *Maxwell 96-238* (CMUB); Chae Son National Park, Pha Ngam Cave, Wang Nuea Distr., partly shaded area along a seasonal stream, mixed evergreen/deciduous hardwood, seasonal forest, rugged limestone area, 500 m alt., immature fr., 27 April 1996 (as *Garcinia* sp.), *Maxwell 96-630* (CMUB); Kamphaeng Phet [Mae Wong National Park, 300–500 m alt., fr., 15 June 1995 (as *G. atroviridis*), *Niyomdham* et al. *4412* (BKF)]; NORTH-EASTERN: Loei [Phu Kradueng National Park, in evergreen jungle, 1200 m alt., fr., 20 August 1954 (as *G. costata*), *Smitinand s.n.* (BKF9460)]; Nakhon Phanom [Phu Lang Ka National Park, Ban Phaeng Distr., in dry evergreen forest, 100 m alt., male fl., 26 February 2007 (*Garcinia* sp.), *Suddee* et al. *3059* (BKF)]; SOUTH-WESTERN: Uthai Thani [Trail to Khao Khiao, Huai Kha Khaeng Wildlife Sanctuary, in dry evergreen forest, fr., 27 July 2002, *Ngernsaengsaruay 141* (BKF spirit collection)]; Kanchanaburi [Khao Yai, East of Sankhla Buri Distr., in dry evergreen forest, rich in bamboo, c. 800 m alt., female fl., fr., 30 March 1968, (as *Garcinia* sp.), *van Beusekom & Phengkhlai 210* (AAU, BKF, C, K, P [05062015]); Chaloem Rattanakosin National Park, Si Sawat Distr., in dry evergreen forest, fr., 27 April 2003, *Ngernsaengsaruay 301* (BKF, QBG spirit collections); Than Rot Noi Cave, Chaloem Rattanakosin National Park, Si Sawat Distr., near stream in mixed deciduous forest, 306 m alt., fr., 27 March 2018, *La-ongsri* et al. *5536* (QBG); ibid., in mixed deciduous forest, 341 m alt., fr., 28 June 2018, *La-ongsri* et al. *5693* (QBG)]; PENINSULAR: Chumphon [Ta Ngao, in evergreen forest, c. 50 m alt., female fl., 20 January 1927 (as *G. cf. planchoni*), *Kerr 11575* (BM, C, K, L [L2417256], P [P04701311]); Lang Suan Distr., in scrub on bank, under 50 m alt., fl., 13 February 1927 (as *Garcinia* sp.), *Kerr 11935* (BM, K)]; Ranong [La-un Distr., in scrub, c. 10 m alt., male fl., 31 December 1928, (as *Garcinia* sp.), *Kerr 16451* (BM, C, K, L [L2409569], P [P05062004])]; Surat Thani [Khlong Sok Subdistr., in evergreen forest, male fl., 12 December 1975 (as *Garcinia* sp.), *Damrongsak 154* (BKF)]; Phangnga [Ban Bang Wan, Tom Nang (cultivated), male fl., 18 December 2004 (as *Garcinia* sp.), *Chamchumroon 2170* (BKF, PSU)]; Krabi [Khao Phanom Bencha National Park, Mueang Krabi Distr., in semi-open area around park office, at edge of lowland evergreen forest, 140 m alt., female fl., 6 December 2004 (as *Garcinia* sp.), *Gardner & Tippayasri ST1195* (K); Khao Phanom Bencha National Park, Mueang Krabi Distr., Thap Prik Subdistr., edge of tropical evergreen rain forest, near headquarters, c. 80 m alt., female fl., y. fr., 14 February 2022, *Ngernsaengsaruay* et al. *G31-14022022* (BKF dry and spirit collections, QBG); Than Bok Khorani National Park, Ao Luek Tai Subdistr., Ao Luek Distr., foot of limestone hill in dry evergreen forest, 20 m alt., male fl., 7 March 2022, *Ngernsaengsaruay* et al. *G32-07032022* (BKF dry and spirit col-

lections, QBG); Ao Nam, Than Bok Khorani National Park, Laem Sak Subdistr., Ao Luek Distr., foot of limestone hill in dry evergreen forest, 15 m alt. female fl. and fr., 7 March 2022, *Ngernsaengsaruay* et al. *G33-07032022* (BKF dry and spirit collections, QBG)]; Nakhon Si Thammarat [Khiri Wong Village, Kam Lon Subdistr., Lan Saka Distr., along the edge of tropical evergreen rain forest, fr., 9 May 2004, *Ngernsaengsaruay 548* (BKF, QBG spirit collections)]; Trang [Na Yong Distr., female fl., 11 March 2018, *Ngernsaengsaruay G24-11032018* (BKF)] (Figure 13).

INDIA. Assam, Goalpara, 10 October 1808, *Wallich Cat. 4860A* (CAL, K-W [K001104 082]); cultivated in H.B.C. (Calcutta Botanical Garden), *Wallich Cat. 4860B* (K-W [K001104 083]); Sylhet, *Wallich Cat. 4860C* (K-W, [K001104084]); H.B.C. (Calcutta Botanical Garden), s.d., *Wight s.n.* (K [K000677592]); H.B.C., East Bengal, s.d., Herb. *Griffith 853* (K [K000677 593]).

BANGLADESH. Sylhet (originally "Silhet" on the label), 1869, *Wallich 18* (P [P04701 875]).

MYANMAR. Chin State, Chin hills, June 1892, *Huk s.n.* (P [P04701874]); Kachin State, Nammun Reserve, Myitkyina Distr., 17 September 1912, *Kyaw 7* (E [E00839549]); Karen State, Mutraw Distr., 10 December 2002, *Htoo 17* (QBG).

VIETNAM. Tonkin, Forêt du Mont Bavi, 1889 (as *G. tonkinensis*), *Balansa 4341* (P [P00 329900, P04701067]); Tonkin, 21 April 1914 (as *G. tonkinensis*), *Unknown 30124* (P [P0489 9645]); Annam, Lang Khoai, 10 June 1924 (as *G. planchoni*), *Poilane 10764* (P [P04701307, P04701310]), Ninh Binh Province, Cuc Phuong National Park, Hoa Binh, 5 August 1999 (as *G. planchonii*), *Cuong 370* (L [L2417255]); Ninh Binh Province, Cuc Phuong National Park, Dang Valley of limestone mountains, 14 March 2000, *Loc* et al. *P10005* (P [P04701876]); Thanh Hoa Province, Cuc Phuong National Park, Thach Thanh Distr., 27 February 2001 (as *G. cowa*), *Cuong & Xinh NMC1298* (P [P04701444]).

LAOS. Khammouan Province, Nakai Distr., 21 May 2006 (as *Garcinia* sp.), *Nanthavong* et al. *BT476* (L [L2409498]); Pak Lai, Expédition du Me-Kong 1866–1868 (as *G. planchoni*), *Thorel s.n.* (P [P04701317]).

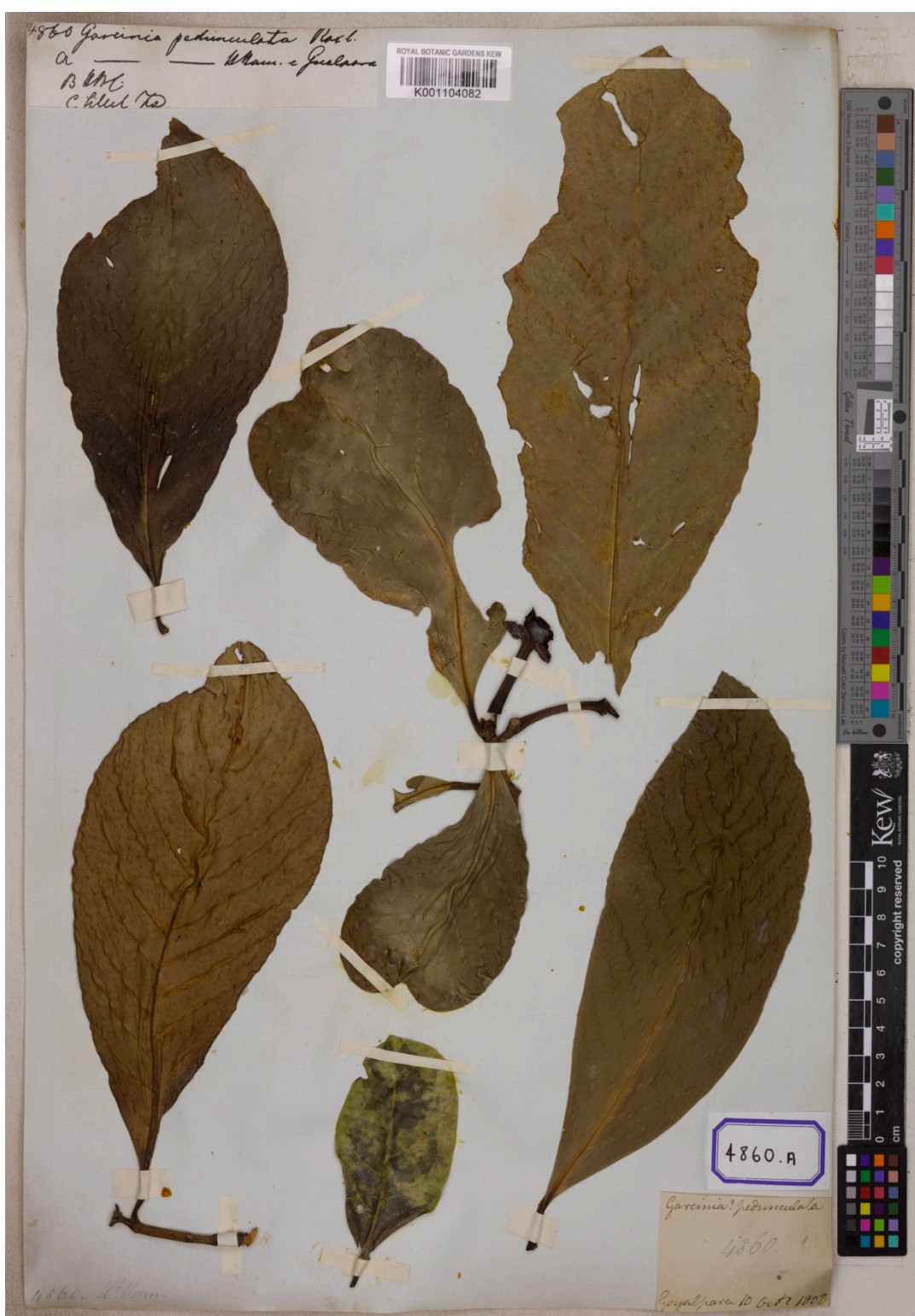

**Figure 9.** Isolectotype of *Garcinia pedunculata*, *Buchanan-Hamilton collection*, *East India Company Herbarium 4860A* (K-W [K001104082!]) from India, Goalpara, with female flowers (http://specimens.kew.org/herbarium/K001104082, accessed on 16 February 2022).

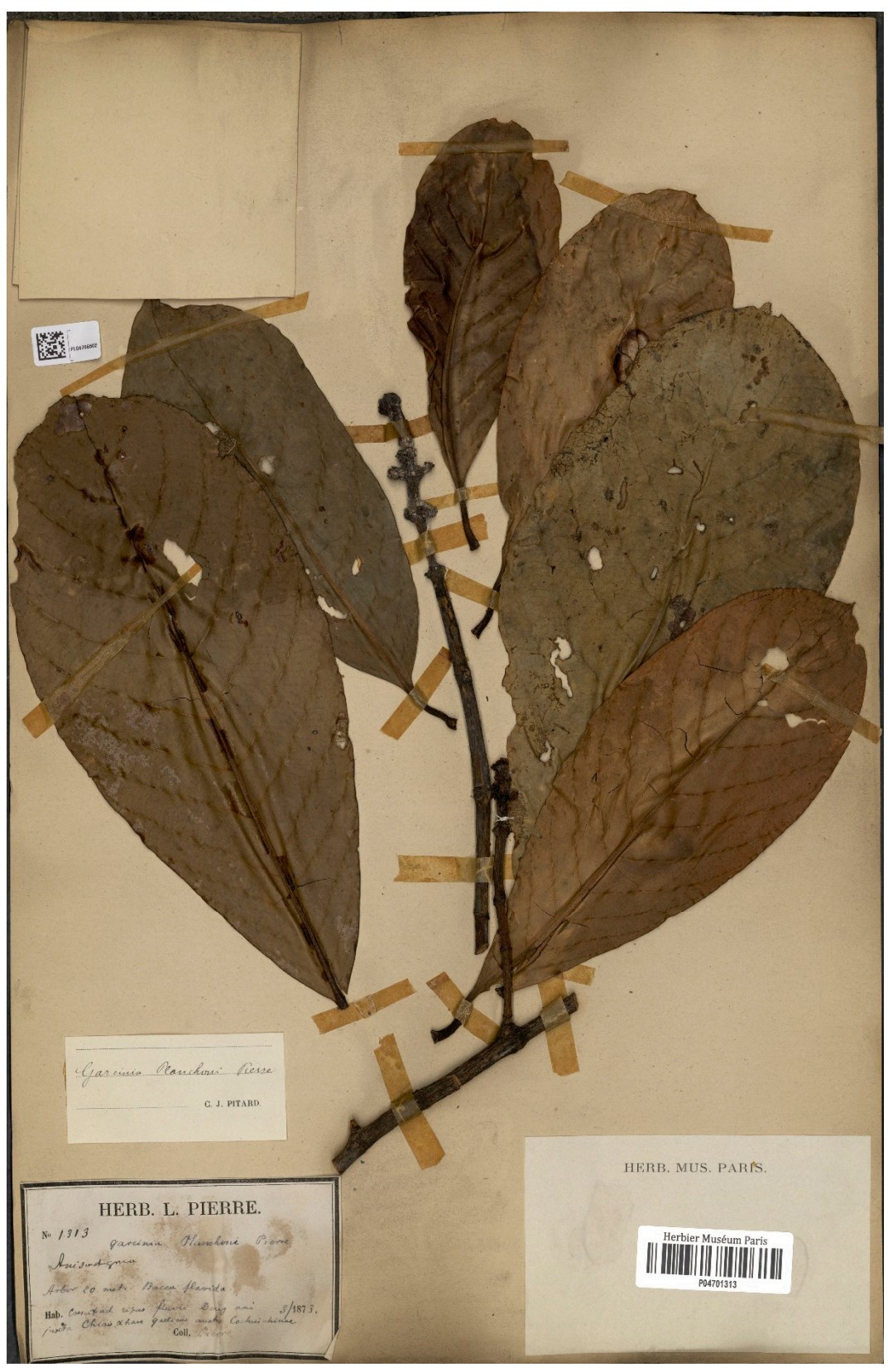

**Figure 10.** Lectotype of *Garcinia planchonii*, *Pierre 1313* (P [P04701313!]) from Vietnam, Crescit ad ripas fluvii Dougnai juxta Chiao Xhan austro Cochinchinae, with female flowers and fruits (http://coldb.mnhn.fr/catalognumber/mnhn/p/p04701313, accessed on 16 February 2022).

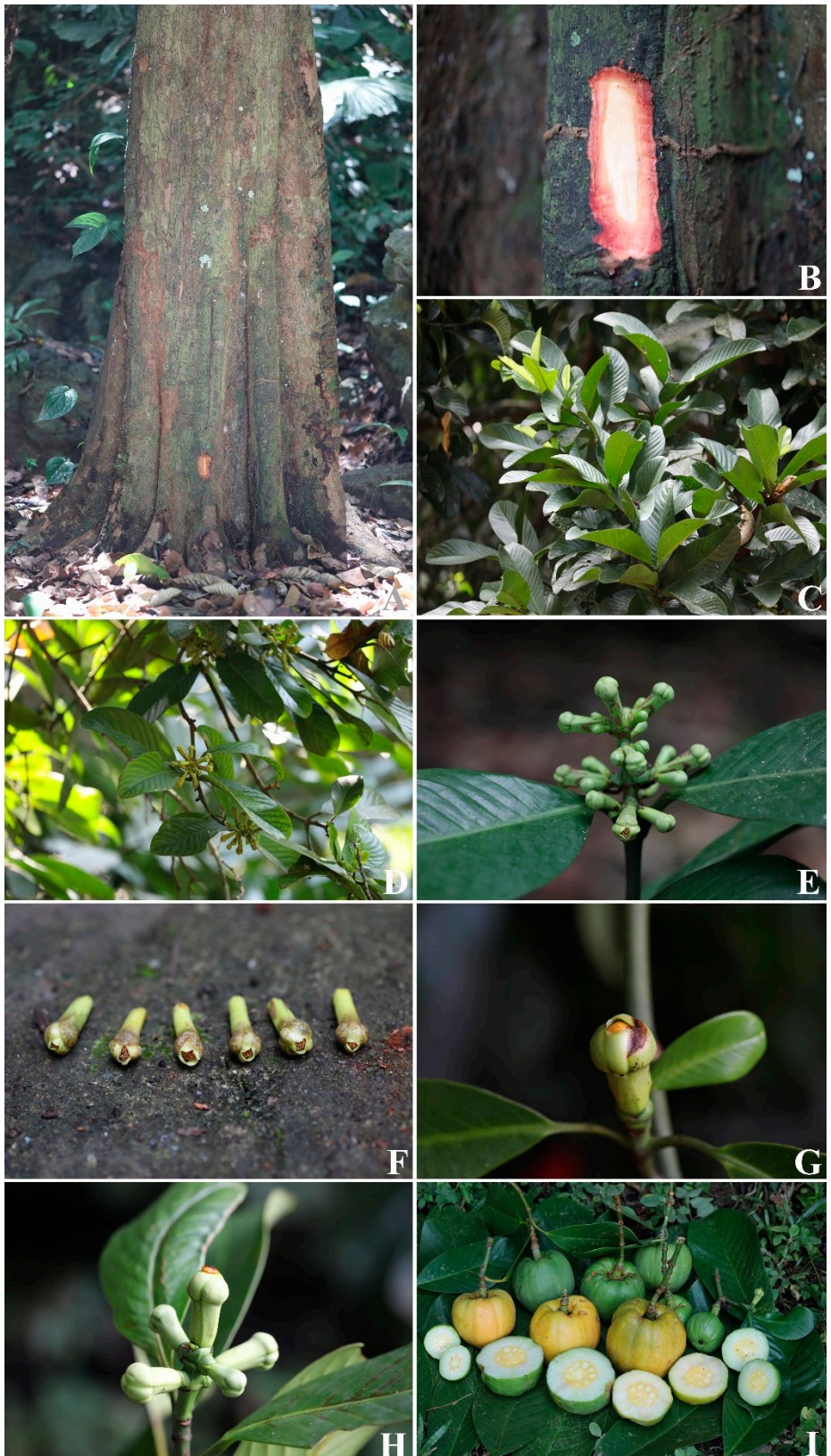

**Figure 11.** *Garcinia pedunculata*: (**A**) stem, narrowly buttressed and outer bark; (**B**) outer bark and inner bark with colourless latex; (**C**) branches, young and mature leaves; (**D**) male flowering branches; (**E**) inflorescence with male flowers; (**F**) fallen male flowers; (**G**) female flower; (**H**) inflorescence

with female flowers; (**I**) mature and ripe fruits, showing pericarps and seeds (cross section). Photos: Chatchai Ngernsaengsaruay.

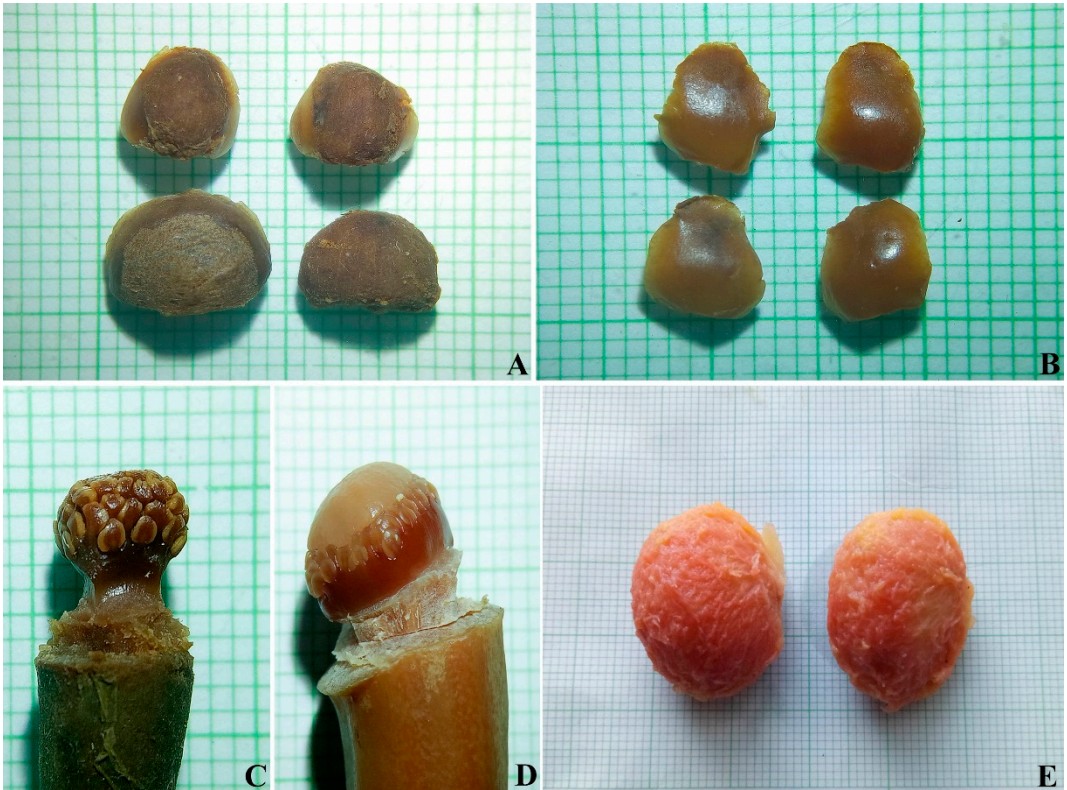

**Figure 12.** *Garcinia pedunculata*: (**A**) outer pair of sepals (lower) and inner pair of sepals (upper); (**B**) outer pair of petals (lower) and inner pair of petals (upper); (**C**) male flower showing stamens numerous, united in a central 4-sided column surrounding a pistillode (sepals and petals removed); (**D**) female flower showing pistil and staminodes (sepals and petals removed); (**E**) seeds. Photos: Weereesa Boonthasak.

*Habitat and Ecology.* It is found in mixed deciduous forests (with or without bamboo), dry evergreen forests, along the edges of dry evergreen and tropical evergreen rain forests, lower montane rain forests and along streams, near above mean sea level up to 1200 m alt.

*IUCN Conservation Status.* *Garcinia pedunculata* is widely distributed from India to Indo-China. It is known from many localities and has a large EOO of 2,443,268.21 km$^2$ and a relatively large AOO of 160 km$^2$. In Thailand, this species is known from the northern, the north-eastern, the south-western and the peninsular regions and has an EOO of 587,997.39 km$^2$ and AOO of 116 km$^2$. Because of this wide distribution and the number of localities, it is considered LC.

*Phenology.* Flowering November to March; fruiting March to August.

*Etymology.* The specific epithet *pedunculata* is a Latin word meaning with the inflorescence supported on a distinct stalk, pedunculate [61,63], for this species refers to inflorescences with distinctly pedicellate male and female flowers or a solitary female flower with a distinct peduncle. The specific epithet of its synonym, *Garcinia planchonii*, honours Jules Émile Planchon (1823–1888), a French botanist.

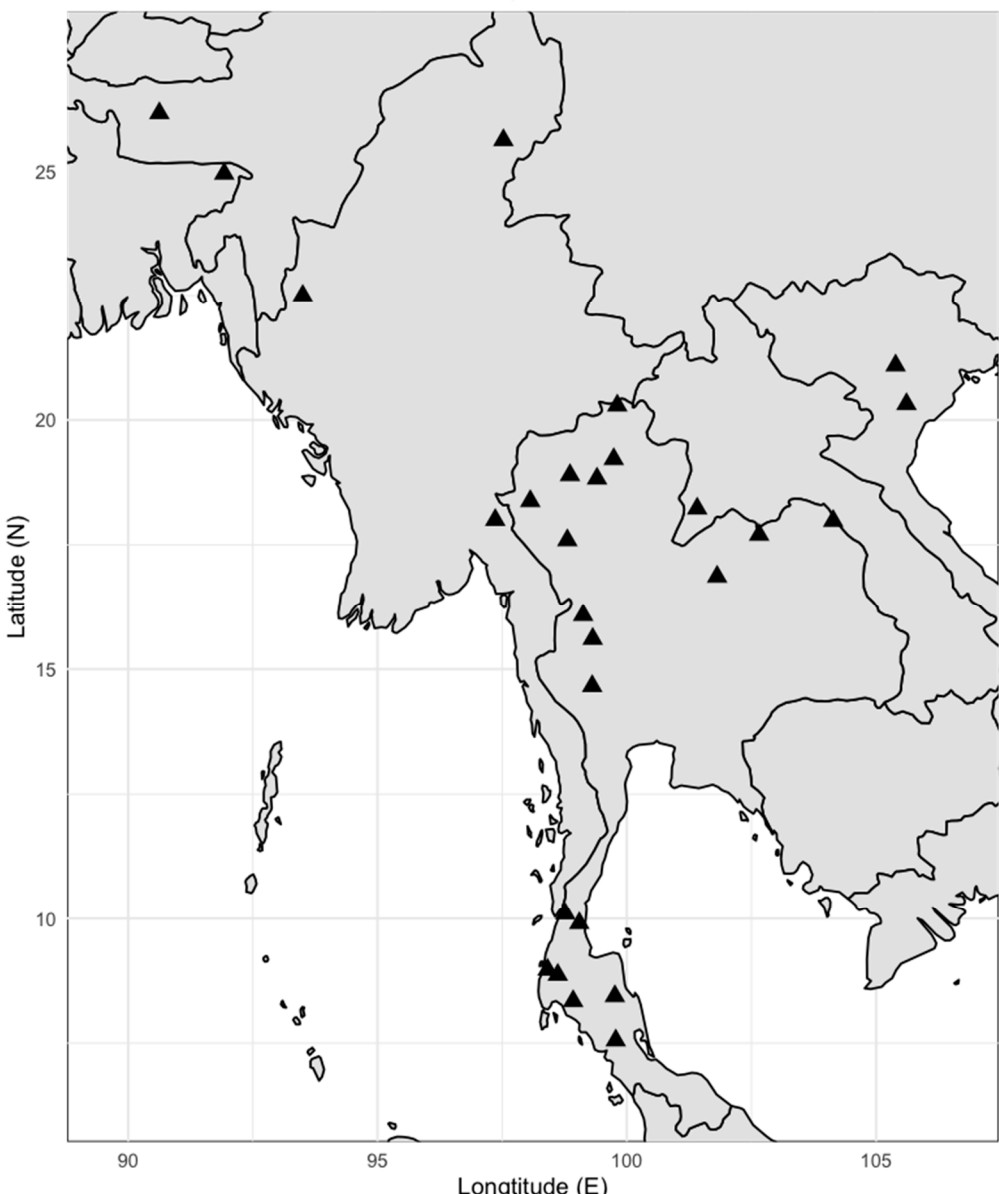

**Figure 13.** Distribution of *Garcinia pedunculata*.

*Vernacular Name*. Krabue chet tua (กระบือเจ็ดตัว) (Chumphon); Ma kwat (มะกวัด) (Lampang); Ma da lot (มะดะหลอด) [19]; Ma nang (มะนั่ง) (Phayao); Ma pong (มะป่อง) (Chiang Rai); Ma ping (มะปิ้ง) (Lamphun); Som chao (ส้มเช้า) (Phangnga); Som mong (ส้มโมง) (Kamphaeng Phet); Som kan dan (ส้มกันดาร) (Nakhon Si Thammarat); **Som khwai** (**ส้มควาย**) (Chumphon, Ranong).

*Uses*. *Garcinia pedunculata* is often cultivated for its fruits. The pericarp, sarcotesta, young shoots and leaves are edible and have a sour taste. Uses of this species are almost the same as *G. atroviridis*. It can be used as a substitute for *G. atroviridis* (the author's observations and interviews). In India and Myanmar, the fruits are eaten raw or cooked, they contain malic acid and are used as a fixative or as a mordant for saffron dye [64–67]. In India, the timber is said to be useful, after seasoning, for making planks, beams and building purposes [64,67].

*Notes*. As for *Garcinia pedunculata*, Buchanan-Hamilton published the name of the species using Roxburgh's name [33], but the original material is all of the material available to Buchanan-Hamilton, possibly including but not limited to any material previously

used by Roxburgh. Anderson mentioned *Wallich Cat. 4860* as the type of *G. pedunculata* [15]. Wallich's collection (East Indian Company Herbarium) numbers are known to be curated by species generally from multiple collections [70]. *Wallich Cat. 4860* represents three gatherings (three different materials collected from three different localities, which are distinguished by A, B and C, respectively). The problem may lie in citing the specimens as *Wallich 4860*. The specimen *4860A* is from India, Goalpara (originally "Gualpara" on the label), a Buchanan-Hamilton collection (a large collection by Francis Buchanan-Hamilton (1762–1829), largely from his survey of the Bengal Presidency (1807–1814), but also from Calcutta Botanic Garden), *4860B* is from India, cultivated in H.B.C. (Calcutta Botanic Garden) and *4860C* is from Sylhet collected by F. De Silva [71]. The various collections under *East Indian Company Herbarium 4860* can only be syntypes if they are original material of the Buchanan-Hamilton protologue. It can easily be argued that Buchanan-Hamilton's own collection is, possibly even the H.B.C. specimen, but it would appear that *4860C* is not original material. Mabberley & Hamilton indicated that the type of *G. pedunculata* is kept at E [43]. I located one sheet of the specimen *Buchanan-Hamilton 1123* at E [E00438017] collected from India, Goalpara (same place, same date and same collector as *4860A*), but it is not a duplicate of *4860A* because it has a different number. Maheshwari has cited "*Wallich 4860*, Goalpara, Bengal (CAL, proposed as neotype)" [7]. It is indeed a mistake, while the original material is available but under the ICN Art. 9.10; this is a correctable error, and Maheshwari has effectively lectotypified the name with *Wallich Cat. 4860A* (CAL). I located one sheet of the cited collection in K-W [K001104082]; therefore, it is an isolectotype.

*Garcinia planchonii* shares essential characters with *G. pedunculata* and is treated here as a new synonym.

The size of the fruits of *Garcinia pedunculata* is 10–18 × 11–20 cm; the fruiting stalk is 5–6 cm long; the ovary has 8–10 locules; the fruits have 8–10 seeds; the shape of the seeds is reniform [16,37], but from my field observations and examined specimens, I found the size of the fruits of this species is 6–10 × 7–12 cm; the fruiting stalk is up to 4 cm long; the ovary has 7–10 locules; the fruits have 5–8 seeds; the shape of the seeds is ellipsoid.

## 4. Discussion

One feature distinguishes *Garcinia* section *Brindonia* from all others, namely, the 4-thecous anthers [22], but I recorded 2-thecous anthers (of 4 pollen sacs) of *G. atroviridis*, *G. lanceifolia* and *G. pedunculata* in this study, in conformity with Gogi & Das [37], Maheshwari [7] and Singh [12]. The pistillode is absent in *Garcinia* section *Brindonia*, except in *G. atroviridis* [22]. In addition, from my study, I found the pistillode in *G. pedunculata*, in conformity with Gogi & Das [37], Li et al. [16], Maheshwari [7] and Singh [12].

*Garcinia* section *Brindonia* is one of the largest and one of the best known sections because several species are cultivated for their edible fruits [40]. In addition, from my study, I found the pericarp (of the fruits), the sarcotesta (of the seeds), the young shoots and leaves and the flowers are edible and have a sour taste, but the flowers are not well known (the author's observations and interviews).

The aril, pulp and pulpy aril are commonly used in *Garcinia* [5,6,12,15,22]. The aril is an outgrowth of the funicle (funiculus), forming an appendage enveloping the seed [72,73], but the sarcotesta is a fleshy layer surrounding the seed and develops from the outer seed coat [72]. I suggest using sarcotesta in this study.

Discussion about lectotypifications, uses and others is mentioned under the species.

The distribution of the three species in *Garcinia* section *Brindonia* in Thailand and surrounding areas are summarised in Table 1. In Thailand, the natural distribution of *G. atroviridis* is confined to the peninsular region (Pattani and Narathiwat Provinces), and this species is also commonly cultivated for its fruits in the same region. It grows mainly in tropical lowland evergreen rain forests, peat swamp forests and along streams, up to 550 m alt. The species is distributed throughout Peninsular Malaysia (Malaya) in lowland forests on the plains and up to 600 m alt. in the mountains. It is extensively cultivated, especially in northern Myanmar [6]. The other two species, *G. lanceifolia* and *G. pedunculata*, have a

wide distribution, each in four Thailand Floristic Regions, but they mostly occur in different habitats and ecology. *G. lanceifolia* is found in the north-eastern, the south-western, the south-eastern and the peninsular regions; the habitat preferences of the species are dry evergreen forests, tropical evergreen rain forests, mixed deciduous forests, along streams in dry evergreen and mixed deciduous forests and limestone, up to 300(–700) m alt. In India, this species is a common undergrowth species in evergreen forests up to c. 900 m. alt. [12]. *G. pedunculata* is known from the northern, the north-eastern, the south-western and the peninsular regions; this species occurs in mixed deciduous forests, dry evergreen forests, along the edges of dry evergreen and tropical evergreen rain forests, lower montane rain forests and along streams, up to 1200 m alt. In India, the species is found in evergreen forests, up to c. 900 m. alt., sometimes cultivated [12].

**Table 1.** Distribution of three species in *Garcinia* section *Brindonia* in Thailand and surrounding areas.

| Species | Thailand Floristic Regions | | | | | | | Surrounding Areas |
|---|---|---|---|---|---|---|---|---|
| | N | NE | E | SW | C | SE | PEN | |
| *G. atroviridis* | | | | | | | × | India (Assam, Arunachal Pradesh), Myanmar, Peninsular Malaysia (Kedah, Penang, Perak, Selangor, Malacca, Johor), Singapore, Indonesia (Sumatra, Borneo) |
| *G. lanceifolia* | | × | | × | | × | × | India (Assam), Bangladesh (Chittagong hills), Myanmar, Vietnam, Laos, Cambodia |
| *G. pedunculata* | × | × | | × | | | × | India (Assam), Bangladesh (Sylhet), Myanmar, China (Yunnan, Tibet), Vietnam, Laos |

## 5. Conclusions

Three names in *Garcinia* section *Brindonia* are lectotypified (*G. gracilis*, *G. lanceifolia* and *G. planchonii*), and a new synonym of *G. pedunculata*, *G. planchonii*, is presented. Three species in this section (*G. atroviridis*, *G. lanceifolia* and *G. pedunculata*) are described and illustrated, along with information on distribution, specimens examined, habitats and ecology, IUCN conservation status, phenology, etymology, vernacular names and uses. The *Garcinia* section *Brindonia* is characterised by stamens in one central mass or column, or in a ring (*Garcinia atroviridis*); anthers 2-thecous (of 4 pollen sacs); stigma generally completely divided into the same number of rays as there are locules of the ovary, papillate; sepals 4 and petals 4; pistillode absent (except in *G. atroviridis* and *G. pedunculata*). This section is one of the best known because several species are cultivated for their edible fruits (e.g., *G. atroviridis*, *G. cowa*, *G. lanceifolia*, *G. pedunculata* and *G. schomburgkiana*). The fruits, young shoots and leaves, and flowers are edible and have a sour taste.

**Author Contributions:** Conceptualization, C.N.; methodology, C.N.; software, C.N.; validation, C.N.; investigation, C.N.; resources, C.N.; data curation, C.N.; writing—original draft preparation, C.N.; writing—review and editing, C.N.; funding acquisition, C.N. All authors have read and agreed to the published version of the manuscript.

**Funding:** This research was funded by the Basic Research Fund (BRF), Faculty of Science, Kasetsart University (2022) and The Carlsberg Foundation, Denmark (2018) managed by Professor Dr Henrik Balslev.

**Institutional Review Board Statement:** Not applicable.

**Data Availability Statement:** Data are available from the corresponding author upon request.

**Acknowledgments:** I would like to thank the Department of Botany and International SciKU Branding (ISB), the Faculty of Science at Kasetsart University in Bangkok for their assistance and use of their facilities. I would like to thank the curators and staff of the following herbaria, AAU, BKF, BM, C, CMUB, K, K-W, P, PSU and QBG, for their assistance during visits and allowing access to the herbarium specimens, and those included in the digital herbarium databases of AAU, BM, E, G,

G-DC, K, K-W, L, U and P. I also would like to thank David Middleton for his kind improvements to the typifications. I am grateful to Prawit Rattanaphan and the staff of Than Bok Khorani National Park, Voradol Chamchumroon head of Khao Hin Son Botanic Gardens and his staff, Sunate Karapan head of Hala-Bala Wildlife Research Station, Prapot Sutthaporn head of Khao Chong Botanical Garden and Janya Jarernrattawong and the staff of Khao Chong Botanical Garden for their facilitation and kind help in the field works. Special thanks go to Weereesa Boonthasak for her kind help with field works and assistance in the laboratory and Nattanon Meeprom for making distribution maps of these three species in *Garcinia*.

**Conflicts of Interest:** The author declares no conflict of interest.

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
