# Peer review of "Lectotypifications of Three Names in Garcinia, Synonymy of Garcinia pedunculata and Detailed Descriptions of Three Species in Garcinia Section Brindonia (Clusiaceae)"

_diversity, doi:10.3390/d14070556_

Round 1

Reviewer 1 Report

The revisions from the first round are good but slight improvement of the English is needed throughout.

Author Response

Response to reviewers and editor

I have carried out corrections suggested by the reviewers and editor.

The corrections as comments of the reviewers, editor and the author are highlighted in red.

The author revised lectotypification of Garcinia pedunculata, therefore the title, abstract and elsewhere must be changed.

Lectotypifications of four names in Garcinia, synonymy of Garcinia pedunculata and detailed descriptions of three species in Garcinia section Brindonia (Clusiaceae)

The author change Figure 9.

Figure 9. Lectotype of Garcinia pedunculata, Buchanan-Hamilton 1123 (lectotype E [E00438017!] from India, Goalpara (https://data.rbge.org.uk/herb/E00438017).

The author added a reference number 34, therefore the former references number 34-72 were changed to 35-73 (see references).

Reviewer 1

Moderate English changes required.

Are all the cited references relevant to the research?

Yes.

Comments and Suggestions for Authors

The revisions from the first round are good but slight improvement of the English is needed throughout.

The author sent the manuscript to SCRIBENDI, Editing and Proofreading Services for academic proofreading.

https://www.scribendi.com/academic.en.html

m/academic.en.html

Reviewer 2 Report

Line 15: A new synonym of G. pedunculata, G. planchonii, is proposed.

Lines 20-21: The results of this study will provide a basis for further development of the other sciences. I am afraid I can t fully understand the meaning of this sentence. Could you please give us 1-2 examples of other sciences?

Line 35: Ngernsaengsaruay & Suddee described a new species of Garcinia from Thailand

Line 247: The fruits are too sour to be eaten raw but taste

Line 290 and elsewhere: following Art. 9.3

Line 458: It is known in many localities ...

Line 735: but the sarcotesta is a fleshy layer surrounding a seed, that develops from the outer seed coat

Line 745: especially in north Myanmar

Author Response

Response to reviewers and editor

I have carried out corrections suggested by the reviewers and editor.

The corrections as comments of the reviewers, editor and the author are highlighted in red.

The author revised lectotypification of Garcinia pedunculata, therefore the title, abstract and elsewhere must be changed.

Lectotypifications of four names in Garcinia, synonymy of Garcinia pedunculata and detailed descriptions of three species in Garcinia section Brindonia (Clusiaceae)

The author change Figure 9.

Figure 9. Lectotype of Garcinia pedunculata, Buchanan-Hamilton 1123 (lectotype E [E00438017!] from India, Goalpara (https://data.rbge.org.uk/herb/E00438017).

The author added a reference number 34, therefore the former references number 34-72 were changed to 35-73 (see references).

Reviewer 2

Moderate English changes required.

The author sent the manuscript to SCRIBENDI, Editing and Proofreading Services for academic proofreading.

https://www.scribendi.com/academic.en.html

Comments and Suggestions for Authors

I have revised the manuscript following the corrections of Reviewer 2 (see the manuscript).

Line 290 and elsewhere: following Art. 9.3

Line 458: It is known in many localities ...

Line 15: A new synonym of G. pedunculata, G. planchonii, is proposed.

Following SCRIBENDI, Editing and Proofreading Services for academic proofreading.

A new synonym for G. pedunculata, namely G. planchonii, is proposed.

Lines 20-21: The results of this study will provide a basis for further development of the other sciences. I am afraid I can’t fully understand the meaning of this sentence. Could you please give us 1-2 examples of other sciences?

In abstract I deleted this sentence “The results of this study will provide a basis for further development of the other sciences, for conservation and sustainable uses of these Garcinia species.”

Abstract: A revision of the genus Garcinia has recently been undertaken by the author as part of the Flora of Thailand. Herbarium specimens deposited in several herbaria, and those included in the digital herbarium databases, were examined by consulting taxonomic literature. In this study, the four names in Garcinia section Brindonia are lectotypified as G. gracilis, G. lanceifolia, G. pedunculata and G. planchonii. A new synonym for G. pedunculata, namely G. planchonii, is proposed. Detailed descriptions, recognitions and illustrations of three species in Garcinia (G. atroviridis, G. lanceifolia and G. pedunculata) are presented, along with information on distributions, specimens examined, habitats and ecology, IUCN conservation status, phenology, etymology, vernacular names and uses. The fruits, the young shoots and leaves, and the flowers of these three species are edible and have a sour taste. These species are often cultivated for their fruits.

Line 35: Ngernsaengsaruay & Suddee described a new species of Garcinia from Thailand

Following SCRIBENDI, Editing and Proofreading Services for academic proofreading.

Ngernsaengsaruay & Suddee described new species, G. nuntasaenii and G. santisukiana, respectively, in Thailand [40,41].

Line 247: The fruits are too sour to be eaten raw but taste

Following SCRIBENDI, Editing and Proofreading Services for academic proofreading.

The fruits are too sour to be eaten raw but are tasty when stewed with sugar [9,57].

Line 745: especially in north Myanmar

, especially in northern Myanmar [72].

Line 735: but the sarcotesta is a fleshy layer surrounding a seed, that develops from the outer seed coat

Following SCRIBENDI, Editing and Proofreading Services for academic proofreading.

, but the sarcotesta is a fleshy layer surrounding the seed and develops from the outer seed coat [5].

Reviewer 3 Report

The author has made little progress with the revision as far as the typifications are concerned. Almost none of my comments have been acted on. The changes made to the manuscript, are in places, incomprehensible e.g. 'Buchanan-Hamilton described a plant, which Roxburgh used Garcinia pedunculata'. The author needs to reconsider the typifications again and stop simply accepting what earlier authors have said, which is mostly wrong.

Author Response

Response to reviewers and editor

I have carried out corrections suggested by the reviewers and editor.

The corrections as comments of the reviewers, editor and the author are highlighted in red.

The author revised lectotypification of Garcinia pedunculata, therefore the title, abstract and elsewhere must be changed.

Lectotypifications of four names in Garcinia, synonymy of Garcinia pedunculata and detailed descriptions of three species in Garcinia section Brindonia (Clusiaceae)

The author change Figure 9.

Figure 9. Lectotype of Garcinia pedunculata, Buchanan-Hamilton 1123 (lectotype E [E00438017!] from India, Goalpara (https://data.rbge.org.uk/herb/E00438017).

The author added a reference number 34, therefore the former references number 34-72 were changed to 35-73 (see references).

Reviewer 3

Extensive editing of English language and style required

The author sent the manuscript to SCRIBENDI, Editing and Proofreading Services for academic proofreading.

https://www.scribendi.com/academic.en.html

Comments and Suggestions for Authors

The author has made little progress with the revision as far as the typifications are concerned. Almost none of my comments have been acted on. The changes made to the manuscript, are in places, incomprehensible e.g. 'Buchanan-Hamilton described a plant, which Roxburgh used Garcinia pedunculata'. The author needs to reconsider the typifications again and stop simply accepting what earlier authors have said, which is mostly wrong.

Garcinia pedunculata is lectotypified here.

Lectotypifications. The protologue of Garcinia pedunculata seems to imply that Buchanan-Hamilton described a new species based on his own specimen but only used the name proposed by Roxburgh [8]. Mabberley & Hamilton (1977) indicated that this type of specimen is kept at E [34]. I located one sheet of the specimen Buchanan-Hamilton 1123 at E [E00438017] collected from India, Goalpara (originally “Gualpara” on the label), but not specified by type, and following Art. 9.6 of the ICN [65], it constitutes a syntype. Therefore, this collection is selected here as the lectotype, following Art. 9.3 and 9.12 of the ICN [65]. Maheshwari referred to G. pedunculata Roxb. ex G. Don (1831) [15], not the Buchanan-Hamilton name [35].

Round 2

Reviewer 3 Report

There has been some improvement this time, but there are still issues:

1. Lectotypification of Garcinia lanceifolia is still based on material that is not original for Roxburgh's name. The specimen is a Wallich collection, NOT a Roxburgh collection.

2. The issue of whether Garcinia pedunculata Roxb. ex Buch.-Ham. is valid or not is still not discussed. Did Buchanan-Hamilton really accept Roxburgh's name for his species? Why is EIC 4860A not considered a duplicate of the Edinburgh specimen? [Same place, same date, same collector] Hamilton seems to have become a coauthor of Mabberley's paper.

3. Opening sentence of Conclusions indicates that names are designated as lectotypes. Lectotypes are designated for names, not the other way round.

Author Response

Response to reviewers and editor

I have carried out corrections suggested by the reviewers and editor.

The corrections as comments of the reviewer 3, editor and the author are highlighted in blue.

The author change Figures 5 and 9.

Figure 5. Lectotype of Garcinia lanceifolia, Unknown collector, East India Company Herbarium 4861B (K-W [K000639523!]) from India, cultivated in H.B.C. (Calcutta Botanical Garden), with female flowers and young fruits (http://specimens.kew.org/herbarium/K000639523).

Figure 9. Isolectotype of Garcinia pedunculata, Buchanan-Hamilton collection, East India Company Herbarium 4860A (K-W [K001104082!]) from India, Goalpara, with female flowers (http://specimens.kew.org/herbarium/K001104082).

Reviewer 3

The author sent the manuscript to SCRIBENDI, Editing and Proofreading Services for academic proofreading. https://www.scribendi.com/academic.en.html

Comments and Suggestions for Authors

There has been some improvement this time, but there are still issues:

  1. Lectotypification of Garcinia lanceifolia is still based on material that is not original for Roxburgh's name. The specimen is a Wallich collection, NOT a Roxburgh collection.

Former comments

For Garcinia lanceifolia Roxb., a lectotype must be a specimen or drawing available to Roxburgh (or a true duplicate of a specimen available to him). A specimen from Calcutta Botanic Garden may be from the plants Roxburgh knew, but if it was gathered after 1815 (when Roxburgh died), it cannot be a lectotype. Wallich was clear in stating which specimens came from Roxburgh's herbarium.

Lectotypifications. Maheshwari cited Wallich Cat. 4861A, 4861B (CAL) from Sylhet as the type of Garcinia lanceifolia [35]. The type material must be from amongst the material cited in Flora Indica 1832. The material for the description must have been collected before 1815 when Roxburgh died because even though it was only published in 1832, the R beside the name means that it is Roxburgh's name and description from his manuscript and not one of Wallich's later additions. It is certainly the case that any material collected by Wallich himself is very unlikely to be original material as he really only became active after Roxburgh had already died. I think this rules out the Bruce collection (H. Bruce is a plant collector for Wallich in Sylhet and Chittagong) in 4861A but the H.B.C. (Calcutta Botanic Garden) collection in 4861B could have been a Roxburgh specimen brought into the East India Company Herbarium. The problem may lie in citing the specimens as Wallich 4861. The specimen 4861B is from H.B.C. collected by an unknown collector at an unknown time, but that it is likely to be original material. It is possible that Roxburgh described it from live plants and didn't make a specimen but I also think it is safe enough to assume this H.B.C. plant could have been from Roxburgh. I located three sheets of the specimen 4861B collected from the same locality: one sheet at K-W [K000639523] and two sheets at P [P04700745, P04700755], and following Art. 9.6 of the ICN [65], they constitute syntypes. Among them, Unknown collector, East India Company Herbarium 4861B (K-W [K000639523]) is in the best condition and clearly shows the diagnostic characters for the species. It is selected here as the lectotype, following Art. 9.3 and 9.12 of the ICN [65].

Type: India, cultivated in H.B.C. (Calcutta Botanic Garden), female fl., y. fr., s.d., Unknown collector, East India Company Herbarium 4861B (lectotype K-W [K000639523!], isolectotypes CAL (not seen), P [P04700745!, P04700755!], designated here)

  1. The issue of whether Garcinia pedunculata Roxb. ex Buch.-Ham. is valid or not is still not discussed. Did Buchanan-Hamilton really accept Roxburgh's name for his species? Why is EIC 4860A not considered a duplicate of the Edinburgh specimen? [Same place, same date, same collector] Hamilton seems to have become a coauthor of Mabberley's paper.

Former comments

Garcinia pedunculata is even more complicated. The first question is whether the Buchanan-Hamilton use of the name is a validation or not. Did Buchanan-Hamilton accept Roxburgh's name for the plant? If you argue he did, then the type has to come from among the specimens of his he refers to as deposited under the name Garcinia pedunculata in India House (these were incorporated by Wallich in the East India Company Herbarium). Maheshwari was referring to G. pedunculata Roxb. ex G.Don (1831) not the Buchanan-Hamilton name.

I cannot see any basis for questioning the validity of the name unless he is suggesting a problem under Art. 36.1. Personally, I don't accept there is a problem. I also disagree with this statement by the reviewer: "the type has to come from among the specimens of his he refers to as deposited under the name Garcinia pedunculata in India House (these were incorporated by Wallich in the East India Company Herbarium)."  What Buchanan-Hamilton says is "The specimens sent to the India-House are marked by the name which Dr Roxburgh used". This means those specimens are original material but the phrase here does not actually exclude the possibility that there is additional Buchanan-Hamilton saw which was not sent to India-House.

Notes. As for Garcinia pedunculata, Buchanan-Hamilton published the name of the species using Roxburgh's name [8] but the original material is all of the material available to Buchanan-Hamilton, possibly including but not limited to any material previously used by Roxburgh. Anderson mentioned Wallich Cat. 4860 as the type of G. pedunculata [3]. Wallich's collection (East Indian Company Herbarium) numbers are known to be curated by species generally from multiple collections [42]. Wallich Cat. 4860 represents three gatherings (three different materials collected from three different localities, which are distinguished by A, B and C, respectively). The problem may lie in citing the specimens as Wallich 4860. The specimen 4860A is from India, Goalpara (originally “Gualpara” on the label), a Buchanan-Hamilton collection (A large collection by Francis Buchanan-Hamilton (1762–1829), largely from his survey of the Bengal Presidency (1807–1814), but also from Calcutta Botanic Garden), 4860B is from India, cultivated in H.B.C. (Calcutta Botanic Garden) and 4860C is from Sylhet collected by F. De Silva [71]. The various collections under East Indian Company Herbarium 4860 can only be syntypes if they are original material of the Buchanan-Hamilton protologue. It can easily be argued that Buchanan-Hamilton's own collection is, possibly even the H.B.C. specimen, but it would appear that 4860C is not original material. Mabberley & Hamilton indicated that the type of G. pedunculata is kept at E [34]. I located one sheet of the specimen Buchanan-Hamilton 1123 at E [E00438017] collected from India, Goalpara (same place, same date and same collector as 4860A), but it is not a duplicate of 4860A because it has a different number. Maheshwari has cited “Wallich 4860, Goalpara, Bengal (CAL, proposed as neotype)” [35]. It is indeed a mistake while original material is available but under the ICN Art. 9.10, this is a correctable error and Maheshwari has effectively lectotypified the name with Wallich Cat. 4860A (CAL). I located one sheet of the cited collection in K-W [K001104082], therefore, it is an isolectotype.

Type: India, Goalpara, female fl., 10 Oct. 1808, Buchanan-Hamilton collection, East India Company Herbarium 4860A (lectotype CAL, isolectotype K-W [K001104082!], designated by Maheshwari [35] (Figure 9).

  1. Opening sentence of Conclusions indicates that names are designated as lectotypes. Lectotypes are designated for names, not the other way round.

Three names in Garcinia section Brindonia are lectotypified (G. gracilis, G. lanceifolia and G. planchonii), and a new synonym of G. pedunculata, G. planchonii, is presented.

This manuscript is a resubmission of an earlier submission. The following is a list of the peer review reports and author responses from that submission.

Round 1

Reviewer 1 Report

This is mostly a straightforward paper which deserves to be published. The descriptions are comprehensive; the illustrations are good. The English is not bad but does need some improvement throughout.

For Garcinia lanceifolia, the author says that the species is not threatened due to its large EOO and AOO. This may be correct but do note that an AOO of 140 km2 is not large and falls within the threshold for Endangered using Criterion B2 if there were few enough locations and also associated threats. This should be clarified. A similar argument is made for G. pedunculata which also does not have a large AOO. LC is undoubtedly correct given the many locations but the author should not state that it has a large AOO.

I think that there is a mistake in the typification of Garcinia pedunculata. The author acknowledges that Wallich 4860A, 4860B and 4860C are all syntypes and, consequently, all suitable for lectotypification. He also noted that Maheshwari neotypified the name with the specimen Wallich 4860A which the author noted was a mistake. It is indeed a mistake while original material is available but under the ICN Art. 9.10, this is a correctable error and Maheshwari has effectively lectotypified the name with Wallich 4860A (CAL). Nevertheless, I suggest this is discussed in this paper rather than remove this taxon from the paper.

Reviewer 2 Report

In the present manuscript, the authors designated the lectotype for five names in Garcinia sect. Brindonia, and reduced a name to the synonymy. It seems to be worthy of publishing. However, the authors does not well understand the methods of lectotypification. Firstly, the authors should analyze what are original material used by the original author of the names when lectotypification. But the authors did not include such discussion in the manuscript. Secondly, the author should discuss why select that specimen as the lectotype from those original materials. The lectotypification in the present paper is mostly based on the images available on the website, the authors did not try to look for all original material in the herbaria. 

When lectotypification, the authors have not searched enough references. Some names has been designated the lectotype by other authors.

(1) Maheshwari (1964) has cited “Grifith, Kew distrih. 862” in K as the type for the name Garcinia atroviridis. Therefore this name has been lectotypified.

(2) Maheshwari (1964) also cited the type for the name Garcinia pedunculata as “Wallich 4860, Goalpara. Bengal (CAL, proposed here as Neotype)”. Here, even though, he cited the Wallich 4860 alone, but he may distinguished the three gatherings by the citation of the locality. The author must discussed the neotypification of Maheshwari if they designated the lectotype for the name. However, I presumed that the authors did not see the publication of Mahesheari (1964).

(3) In the protologue of Garcinia pedunculata, it seems to imply that Buchan-Hamilton described the new species based on his own specimen but only used the name proposed by Roxburgh.. Mabberley & Hamilton (1977) indicated that such specimens are kept at E. The authors might clarify this problem.

Reviewer 3 Report

Dear authors please find a few comments and corrections

Line 31: that the Malay Peninsula

Line 35: new species of Garcinia

Line 62: during the fieldwork.

Line 66: of the Extent of Occurrence

Line 98: reddish-green

Line 135: greenish-brown.

Line 145:  in an evergreen forest by a stream

Line 159: in an evergreen rain forest

Line 225: We, therefore, 

Line 236: Garcinia atroviridis is commonly cultivated 

Line 252: are applied to women after 

Line 312: with inter secondary veins,

Line 326: longitudinally dehiscent; 

Line 330: basally united in several bundles surrounding the base of the ovary 

Line 400: in an evergreen forest by a stream,

Lines 404-405: in a mixed deciduous forest by a stream,

Line 558: longitudinally dehiscent;

Line 564:  basally united in several bundles surrounding the base of the ovary

Line 733: a fleshy layer surrounding a seed

Reviewer 4 Report

There are problems with some of the lectotypifications presented in the paper. The author does not provide a thorough analysis of the protologue for the names concerned and the original material available. 

For Garcinia lanceifolia Roxb., a lectotype must be a specimen or drawing available to Roxburgh (or a true duplicate of a specimen available to him). A specimen from Calcutta Botanic Garden may be from the plants Roxburgh knew, but if it was gathered after 1815 (when Roxburgh died), it cannot be a lectotype. Wallich was clear in stating which specimens came from Roxburgh's herbarium.

Garcinia pedunculata is even more complicated. The first question is whether the Buchanan-Hamilton use of the name is a validation or not. Did Buchanan-Hamilton accept Roxburgh's name for the plant? If you argue he did, then the type has to come from among the specimens of his he refers to as deposited under the name Garcinia pedunculata in India House (these were incorporated by Wallich in the East India Company Herbarium). Maheshwari was referring to G. pedunculata Roxb. ex G.Don (1831) not the Buchanan-Hamilton name.

For Garcinia atroviridis, the 862 number is not Griffith's number, it is the number put on by Hooker when he sorted the material for distribution well after Griffith's death.